



# Multi-day hail clusters and isolated hail days in Switzerland – large-scale flow conditions and precursors

Hélène Barras[1,2], Olivia Martius[1], Luca Nisi[2], Katharina Schroeer[3], Alessandro Hering[2], and Urs Germann[2]

[1]Mobiliar Lab for Natural Risks, Institute of Geography, Oeschger Centre for Climate Change Reserach, University of Bern, Bern, Switzerland
[2]Federal Office of Meteorology and Climatology, MeteoSwiss, Locarno-Monti, Switzerland
[3]Federal Office of Meteorology and Climatology, MeteoSwiss, Zurich, Switzerland

**Correspondence:** Hélène Barras (helene.barras@giub.unibe.ch)

**Abstract.** In Switzerland, hail regularly occurs in multi-day hail clusters. The atmospheric conditions prior to and during multi-day hail clusters are described and contrasted to the conditions prior to and during isolated hail days. The analysis focuses on hail days that occurred between April and September 2002—2019 within 140 km of the Swiss radar network. Hail days north and south of the Alps are defined using a minimum area threshold of a radar-based hail product. Multi-day

clusters are defined as 5-day windows containing 4 or 5 hail days and isolated hail days as 5-day windows containing a single hail day. The reanalysis ERA-5 is used to study the large-scale flow in combination with objectively identified cold fronts, atmospheric blocking events, and a weather type classification. Both north and south of the Alps, isolated hail days have frequency maxima in May and August-September whereas clustered hail days occur mostly in July and August. Composites of atmospheric variables indicate a more stationary and meridionally amplified atmospheric flow both north and south of the

Alps during multi-day hail clusters. On clustered hail days north of the Alps, blocks are more frequent over the North Sea, and surface fronts are located farther from Switzerland than on isolated hail days. Clustered hail days north of the Alps are also characterized by significantly higher convective available potential energy (CAPE) values, warmer daily maximum surface temperatures, and higher atmospheric moisture content than isolated hail days. Hence, both stationary flow conditions and anomalous amounts of moisture are necessary for multi-day hail clusters on the north side. In contrast, differences in CAPE

on the south side between clustered hail days and isolated hail days are small. The mean sea level pressure south of the Alps is significantly deeper, the maximum temperature is colder, and local moisture is significantly lower on isolated hail days. Both north and south of the Alps, the upper-level atmospheric flow over the eastern Atlantic is meridionally more amplified three days prior to clustered hail days than prior to isolated days. Moreover, Moreover blocking occurs prior to more than 10 % of clustered hail days over Scandinavia, but no blocks occur prior to isolated hail days. Half of the clustered hail days south of the

Alps are also clustered north of the Alps. On hail days clustering only south of the Alps, fronts are more frequently located on the Alpine ridge, and local low- level winds are stronger. The temporal clustering of hail days is coupled to specific synoptic- and local- scale flow conditions, this information may be exploited for short to medium-range forecasts of hail in Switzerland.



## 1 Introduction

In Switzerland, hail days can occur several days in a row. Such multi-day clusters of hail days can cause substantial damage
in a short time. Multi-day clusters of severe weather and associated high impacts have also been reported from North America
(Shafer, C., Doswell III, 2012; Trapp, 2014; Schroder and Elsner, 2020; Gensini et al., 2019). Although the atmospheric con-
ditions associated with hail in Switzerland and central Europe are well studied (e.g., Huntrieser et al., 1997; Madonna et al.,
2018; Taszarek et al., 2017; Púčik et al., 2015; Brooks, 2009; Púčik et al., 01 Nov. 2019; Kunz et al., 2020), little is known
about the large-scale weather conditions that lead to multi-day clusters of hail days. Such multi-day clusters are likely the result
of the extended longevity or repeated re-establishment of particular features of weather situations over Europe. Addressing this
research gap is relevant for insurance and forecasting applications. For insurance companies, an important question is whether
hail events can be considered as independent or not. For forecasting, it is relevant to know whether processes and weather
situations leading to isolated hail events and multi-day clusters of hail events differ substantially from each other.

Large-scale flow patterns have been linked to the spatial and annual variability of thunderstorms in Europe (Piper et al.,
2019; Mohr et al., 2019), and two case studies highlight two particularly long-lasting sequences of consecutive thunderstorm
days in central Europe (Piper et al., 2016; Mohr et al., 2020). Piper et al. (2016) compare a 15-day episode of thunderstorms
in Germany in May–June 2016 with the period 1960–2014 and find that this event was exceptional for its number of days with
prevailing extreme precipitation or convection-favoring conditions. Mohr et al. (2020) investigate a series of severe thunder-
storms in May–June 2018 in central Europe and find a blocking anticyclone that trapped moist and warm air over western and
central Europe and several cut-offs on the block's southern fringe to provide exceptionally persistent low-stability conditions.
Madonna et al. (2018) compared large-scale conditions during June 2006, a month with above-average hail days (12, average
9.2) with June 2004, where only 2 hail days occurred in northern Switzerland. June 2006 saw warmer surface temperatures over
most of Europe, higher CAPE values, more moisture, and more unstable local conditions than the climatology. During June
2004, reanalysis data indicates increased blocking frequency south and west of Greenland, less moisture, and more frequent
lows and fronts in the Alpine region and north of the Alps. Whereas these investigations have studied individual cases in detail,
an analysis of the synoptic and large-scale conditions during and leading to multi-day hail clusters in Switzerland has yet to be
conducted.

The first objective of this study is to quantify the occurrence of multi-day hail clusters in Switzerland and northern Italy in
the period from 2002–2019. The second objective is to identify the main features of large-scale circulation over Europe during
and prior to multi-day hail clusters and contrast these with those of the circulation on isolated hail days.

More specifically we aim to answer two questions:

– Which atmospheric conditions are associated with and differentiate multi-day hail clusters and isolated hail days in
Switzerland north and south of the Alps during 2002—2019?



– Which atmospheric conditions occur on days before multi-day and isolated hail events?

The paper is structured as follows: Sect. 2 presents the data we use in this study. Sect. 3 focuses on the methods: how we defined clustered and isolated hail days and the method for determining the statistical significance of the difference between composites. Sect. 4 describes the results, which are discussed and summarized in Sect. 5. The article ends with the conclusions and outlook in Sect. 6.

## 2 Data

### 2.1 Probability of hail (POH)

This study uses the radar- and model-based probability of hail product (POH, Foote et al., 2005 based on Waldvogel et al., 1979) to identify hail days between April and September 2002—2019 in the Swiss radar domain. POH is an operational product that indicates the likelihood of hail at the ground (zero to 100 %) on a 1 x 1 km Cartesian grid, with radar hail data quality generally assumed to be highest within a 160 km radius around the five Swiss weather radar stations (Nisi et al., 2016). Car insurance loss data has verified a threshold of POH $\geq$ 80 % to indicate the presence of hail locally (Nisi et al., 2016; Madonna et al., 2018). The extent of the daily area of POH $\geq$ 80 % is extracted separately for a domain north of the Alps (Fig. 1, blue area) and another south of the Alps (Fig. 1 green area). The domain south of the main Alpine ridge contains southern Switzerland and a region of Northern Italy within a 140 km radius of the weather radar stations (Fig. 1).

### 2.2 Car insurance loss reports

Area thresholds for the identification of hail days are defined with hail-related car insurance loss reports provided by the Swiss Mobiliar insurance company. The insurance loss reports are available for the years 2003—2012 and are described in detail in Morel (2014) and Nisi et al. (2016). Morel (2014) shows that some car insurance loss dates had to be corrected because of human error. To increase the robustness of the car insurance loss information, we consider only days with at least five car insurance loss reports.

### 2.3 Weather Type Classification

This study uses an automatic daily weather type classification (WTC) of the synoptic situation over Central Europe (Weusthoff, 2011). The WTC has ten classes: eight classes for the eight main wind directions and two classes for low- and high-pressure situations based on the geopotential height at 500 hPa. The wind and geopotential data are taken from the ERA-Interim reanalysis.

### 2.4 Reanalyses

The two reanalysis data sets used in this study (ERA-Interim, see Dee et al., 2011, and ERA-5, see Hersbach et al., 2020) are produced by the European Centre for Medium-Range Weather Forecasts (ECMWF). Our analysis considers the period April

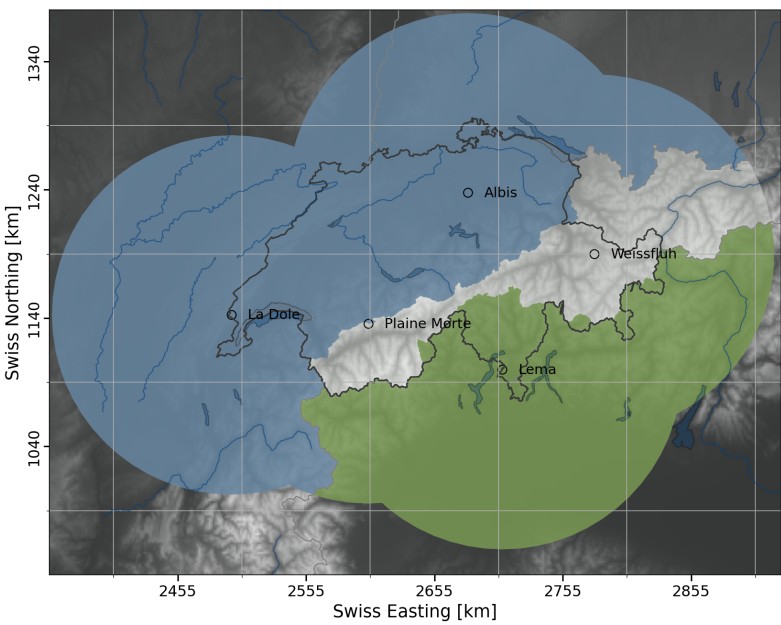

**Figure 1.** The investigation areas south of the Alps (green) and north of the Alps (blue) overlaid on a topographical map (gray shading). Light grey shading indicates the area within a 140 km radius of the five Swiss weather radar stations with the highest radar quality (Nisi et al., 2016). Switzerland is centered at 46.8°N 8.2°E.

2002–September 2019. We extracted ERA-5 variables at a 6-hourly temporal resolution and a spatial resolution of 0.5°. The large-scale dynamics are described through the Ertel potential vorticity (PV; in PV units (PVU)), calculated and interpolated to the 335 K isentrope, and horizontal wind components at 250 hPa (in m s$^{-1}$). Daily atmospheric blocking events were calculated as in Rohrer et al. (2018), following the algorithm developed by Schwierz et al. (2004). Synoptic and local conditions are
represented by low-level winds (at 850hPa, in m s$^{-1}$), the daily maximum surface temperature (T2M, in degrees Celsius), the daily maximum convective available potential energy (CAPE, in J kg$^{-1}$) and the daily mean sea-level pressure (MSLP, in hPa). The daily statistics for these last three variables are calculated from hourly values. Bulk wind shear values are obtained by subtracting the horizontal wind components at 850 hPa from the wind at 500 hPa. The total precipitable water (TPW, in mm) provides information on the moisture content of the atmosphere. The front data stem from the ERA-Interim reanalysis that has
been interpolated to a spatial grid of 1°and has a temporal resolution of 6 hours (see Schemm et al., 2015, for details). These fronts have a minimum gradient of equivalent potential temperature of at least 4 K per 100 km at 850 hPa and a minimum length of 500 km. The composites show the percentage of all time steps with fronts.





## 3   Methods

### 3.1   Definition of hail days

We identify hail days by denoting the area where POH equals or exceeds 80 % during a day as the daily POH footprint. To determine hail days, we need to define a minimum footprint area. This is because, despite rigorous data quality control in the Swiss operational radar data processing, some data points still have residual radar artefacts not related to hail. The number of data points affected is small considering the amount of ground clutter in the raw radar data for an Alpine country, but we have to take them into account when identifying hail days using daily POH footprints. We tested minimum footprint area thresholds

between the 70th and the 95th percentile of the daily footprint area distribution in the northern and southern domains. We found that the 80th percentile of the area distribution is best suited to identifying hail days. This threshold best corresponds to days with car damage reported across Switzerland over 2003–2012 (Table A1 in the Appendix). If we use the 80th percentile to define hail days, most days with $\geq 5$ car insurance losses occur on hail days, and the number of days with $\geq 5$ car insurance losses occurring on nonhail days are minimized. As a result, we define a hail day as a day with a footprint greater than the 80th

percentile of the POH $\geq 80$ % area distribution. This corresponds to a daily maximum POH $\geq 80$ % over an area greater than 580 km$^2$ in the northern domain and greater than 499 km$^2$ in the southern domain.

This definition produces an average of 26 hail days per hail season north of the Alps; a minimum of 16 hail days occurred in 2014 and a maximum of 43 hail days in 2009. South of the Alps, it produces an average of 25 hail days per hail season; a

minimum of 15 hail days occurred in 2004 and 2007 and a maximum of 38 hail days in 2019.

### 3.2   Selection of serially clustered versus isolated hail days

To define the clustered hail periods and isolated hail days, we use a counting approach similar to Pinto et al. (2014) and Kopp et al. (2021). In a period of 5 days we require multi-day clustered hail periods to have at least 4 hail days and isolated hail periods to have only 1 hail day. To ensure independence, all isolated hail days must have a period of at least 3 nonhail days to

the next hail day.

All hail days in 2002–2019 and their assignment to the clustered or isolated hail day category are shown in Fig. 2 and counted in Table 1. In total, 308 hail days are identified north of the Alps and 294 hail days south of the Alps. North of the Alps, the period with clustered hail days starts on day of the year (DOY) 129 (mid-May) and ends on DOY 238 (end of August). South of

the Alps, the period with clustered hail days starts a month later on DOY 160 (mid-June) and ends on DOY 230 (mid-August). Isolated hail days occur earlier and later in the season than clustered hail days (Fig. 2). To avoid seasonality effects, all isolated hail days outside the seasonal range of the clustered hail days are excluded, leaving 69 isolated hail days north of the Alps and 42 isolated hail days south of the Alps. For significance testing, we separate the clustered hail days into independent clustering periods of 5 days. We define independent periods as 5-day periods separated by at least 2 days. In addition, series of hail days

that cluster for more than 11 days, for example in 2003, are split into independent 5-day clustering periods that each contain

 

at least 4 hail days. This results in 32 independent 5-day periods with a total of 135 clustered hail days north of the Alps and 21 of these 5-day periods with 89 hail days south of the Alps. About half of all clustered hail days south of the Alps are also clustered hail days north of the Alps (Fig.2). In summary, this article analyzes 204 (135+69) of a total of 308 hail days north of the Alps and 131 (89+42) of 294 hail days south of the Alps (Table 1).

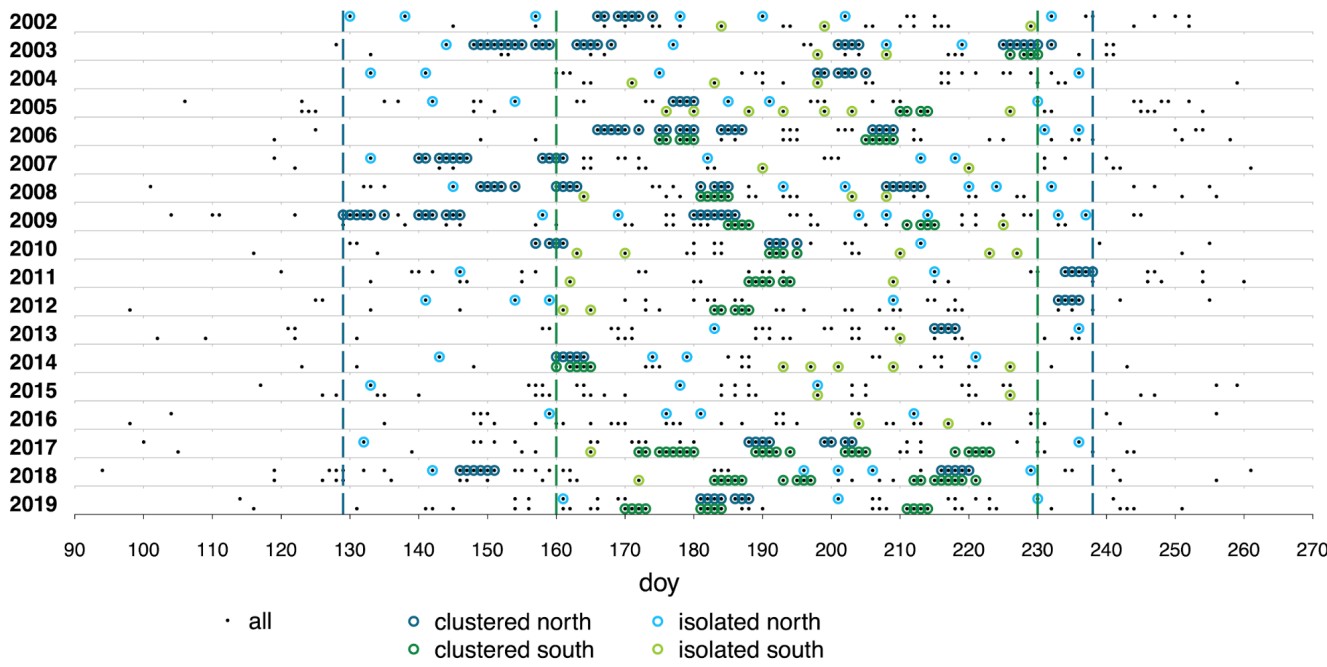

**Figure 2.** All hail days for each year between 2002 and 2019, north of the Alps (top rows, blue colors) and south of the Alps (bottom rows, green colors); the clustered hail days have dark circles, and the isolated hail days have light colored circles. Note that isolated hail days are only considered in the analysis if they occur no earlier or later in the year than the earliest or latest clustered hail day (marked by vertical dashed lines).

### 3.3 Composites of large-scale flow

Differences in large-scale conditions during clustered and isolated hail days are assessed using composites of the large-scale flow from reanalysis data. The composites are built for all 135 clustered and 69 isolated hail days north of the Alps and for all 89 clustered and 42 isolated hail days south of the Alps (Table 1). We further build composites of the atmospheric circulation prior to a hail event on the first (d-1), second (d-2), and third (d-3) nonhail days. For clustered hail events, the first hail day of the cluster is day 0. For clustered hail events north of the Alps (south of the Alps), d-1, d-2, and d-3 each include 31 (21) days. North of the Alps, the number of days is not 32, the number of independent clusters, because in 2003, two independent clustering periods shared a d-1, d-2, and d-3 day. For isolated hail events, north of the Alps we find 69 days and south of the Alps 42 days for d-1, d-2, and d-3 (see Table 1). Because half of the clustered hail days south of the Alps are also clustered





**Table 1.** Number of hail days north and south of the Alps and the number of days leading to them. The bold columns indicate how many days enter the composites of the large-scale flow.

| | | clustered events | | | isolated events | | |
|---|---|---|---|---|---|---|---|
| | total number of hail days | clustered hail days | **days before clustered hail days (d-1, d-2, d-3)** | **clustered hail days in independent 5-day periods** | isolated hail days | **days before isolated hail days (d-1, d-2, d-3)** | **isolated hail days in the DOY range of clustered hail days** |
| North of the Alps | 308 | 164 | **31** | **135** | 96 | **69** | **69** |
| South of the Alps | 294 | 102 | **21** | **89** | 99 | **42** | **42** |

north of the Alps, we also created composites comparing hail days that are clustered both south and north of the Alps and
compare them to hail days that are only clustered south of the Alps.

### 3.4   Calculating the statistical significance of the differences

We apply two-sample Kolmogorov-Smirnov (KS) tests (Bonamente, 2017, p.219-221) on 500 series of hail days. These were
resampled from the original set of hail days to infer the statistical significance of the differences between composites of the
atmospheric variables during clustered and isolated hail days. More details on the resampling method are available in the
Appendix Sect. B. By applying the KS-test to 500 resample series, each difference between isolated and clustered hail day
composites has 500 significance test results. To reduce the chance of type I errors in the large number of p-values, we control
the probability of rejecting the null hypothesis with the false discovery rate and limit the probability that a rejected null
hypothesis should have been accepted to $\alpha_{FDR}$ = 10 % (Wilks, 2016). In the difference maps presented in the results section,
we highlight the areas where more than 50 % or 80 % of the 500 tests indicate significant differences. The binary variables,
fronts and blocks, are not tested for significance, and differences are only discussed qualitatively.

### 4   Results

### 4.1   Seasonality of isolated and clustered hail days

The seasonality of all hail days, the clustered hail days, and the isolated hail days are illustrated by showing the total number of
hail days per 20-day window across the hail season in the period from 2002 to 2019 (Fig. 3). On both sides of the Alps, isolated
hail days occur earlier and later in the year than clustered hail days. Clustered hail days occur earlier in the hail season north
of the Alps than south of the Alps. South of the Alps, clustered hail days exhibit a pronounced peak in the middle of the year,

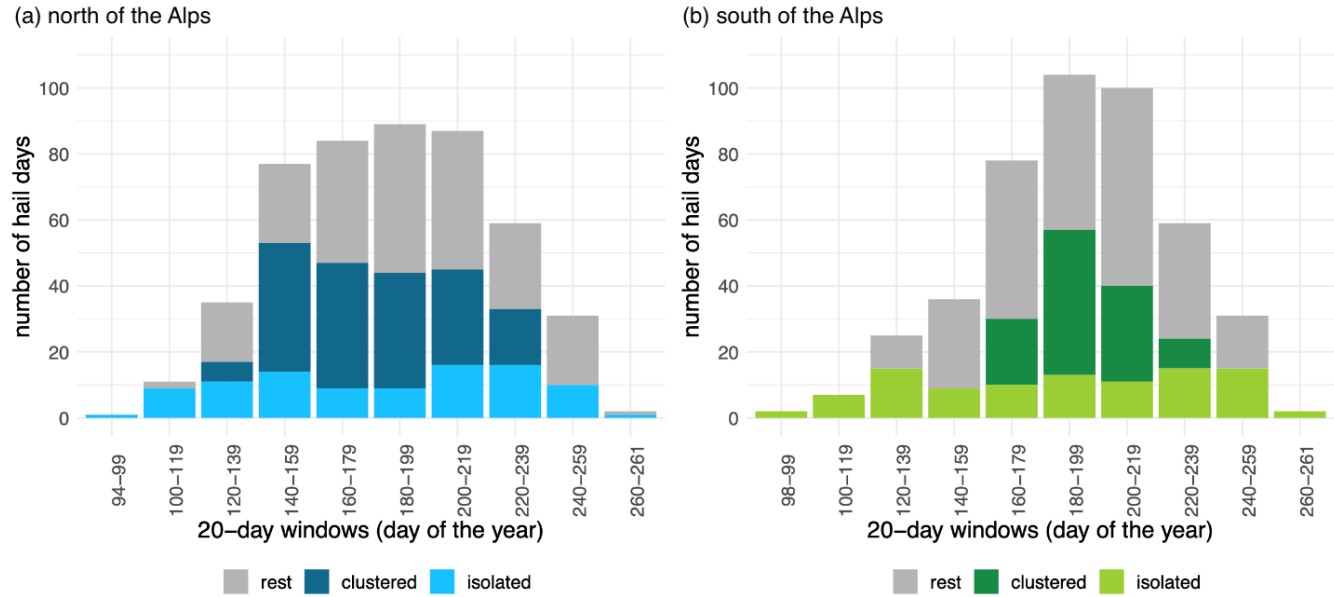

**Figure 3.** The total number of hail days (grey bars), the number of clustered hail days (darker color), and the number of isolated hail days (lighter color) for 20-day windows across the hail season for hail events north of the Alps (a, blue colors) and south of the Alps (b, green colors) between 2002–2019. The numbers on the x-axis indicate the day of the year. In this graph, the isolated hail days are also shown for the period outside of which clustered hail days are defined.

whereas isolated hail days are distributed more uniformly across the hail season. There is a notable year-to-year variability (see Fig. 2). Both north and south of the Alps, 20-day periods without clustered or isolated hail days occur in at least one of the 18 years.

## 4.2 Weather type classifications

We start by comparing the distribution of central European weather types during clustered and isolated hail days. For all classes of hail days, the two most common weather types are westerly (W) and southwesterly (SW) winds in central Europe (Fig. 4). North of the Alps, westerly flow is more common during isolated hail days (43 % compared to 27 %), and northerly flow is more common during clustered hail days (22 % compared to 10 %). South of the Alps, the fraction of days with a northwesterly flow is about twice as frequent (8 % higher) for clustered hail days than for isolated ones. However, the overall similarity in fraction of weather types for all classes of hail days suggests that the main wind direction over central Europe is too general an indicator to differentiate between clustered and isolated hail days, and we therefore present more detailed composites of the large-scale flow next.





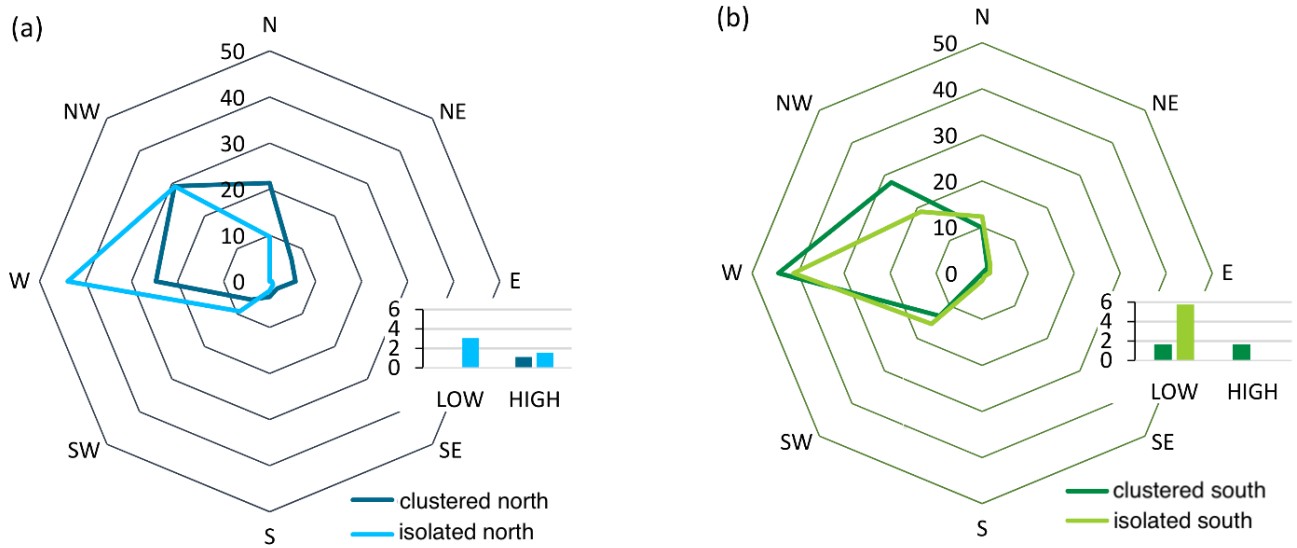

**Figure 4.** Relative frequency [%] of weather types per hail day category for hail events a) north of the Alps and b) south of the Alps. The eight directional weather types are shown in the spider plot, and low- and high-pressure weather type frequencies are indicated in the inset.

## 4.3 Large-scale weather situation during clustered and isolated hail days

### 4.3.1 Atmospheric conditions over Europe and the North Atlantic on hail days north of the Alps

The upper-level atmospheric flow is represented by composites of the PV on the 335 K isentrope. In the PV composite for the clustered hail days, a trough is located over western Europe with its axis at 10°W (Fig. 5a). The trough extends meridionally to southern Spain. A downstream ridge with its axis located at 12°E over Italy tilts anticyclonically. Downstream of the ridge, an anticyclonically tilted trough extends from the Black Sea to the eastern Mediterranean. The anticyclonic tilt of the ridge and

the trough point to anticyclonic Rossby wave breaking over central and eastern Europe. During at least 5-10% of clustered hail days, an atmospheric block is located at 66°N between Scandinavia and Iceland.

In the PV composite for isolated hail days, a trough is present over western Europe with the trough axis located at 2°W (Fig. 5c). The meridional amplification of this trough is slightly weaker than the western European trough during the clustered

hail days (Fig. 5e), but the trough on isolated days is deeper at 50°N. The axis of the downstream ridge is located at 12°E (Fig. 5c). The ridge does not exhibit a noticeable tilt. A downstream trough is located over the Black Sea. The meridional amplitude of this trough is smaller than that of its counterpart in the clustered hail day composite (Fig. 5e). Zonal winds at 250 hPa are significantly weaker over the Mediterranean and central Europe (Fig. 5e). In the isolated hail days composite, a stronger ridge is located upstream of Europe over the Atlantic at 30°W and a trough at 55°W (Fig. 5a, c and e). During at least

5-10% of the isolated hail days, atmospheric blocking occurred over North America at 60-70°N and 50-80°W. The jet over





the central Atlantic has a southwest– northeast tilt during isolated hail days. Hence, the upper-level flow during clustered hail days is characterized by a longer wavelength of the waves over Europe, by a stronger meridional amplification of the troughs, by wave breaking, and by a weaker zonal flow over Europe than during the isolated hail days. All of these factors indicate a more stationary flow situation over Europe during the clustered hail periods.


The air contains significantly more moisture ( 3-5 mm) over western and central Europe north of the Alps on clustered hail days (Fig. 5b, d and f). On clustered hail days, the winds are on average weaker than 4m s$^{-1}$ at 850hPa and flow from SSW over northern Switzerland (Fig. 5b). On isolated hail days, the winds at 850hPa are significantly stronger and southwesterly (Fig. 5d and f).

On clustered hail days, maximum daily temperatures are significantly warmer than isolated hail days (+1 to +4 K) over northern Switzerland (Fig. 6a). Furthermore, the sea level pressure is significantly higher than on clustered than on isolated days over Central Europe and northern Europe (+2 to >+5 hPa, Fig. 6e). The sea-level pressure pattern and the weaker lower tropospheric zonal winds north of the Alps on clustered hail days (Fig. 5b) indicate more stationary conditions north of the Alps. No significant differences in the wind shear are found over Switzerland (not shown).


On clustered hail days, cold fronts are less frequent just north of the Alps and over central France than on isolated hail days (Fig. 6b and d) and more frequent over northern France and the English Channel. Hence, cold fronts are further away from northern Switzerland during the clustered hail days.

On clustered hail days, CAPE values over central Europe and the Mediterranean are larger than on isolated hail days (Fig. 6b and d). The difference over northern Switzerland is significant and substantial at >400 J kg$^{-1}$ (Fig. 6f). In summary, persistent warmer and more humid conditions north of the Alps during clustered hail days contribute to significantly higher instability.

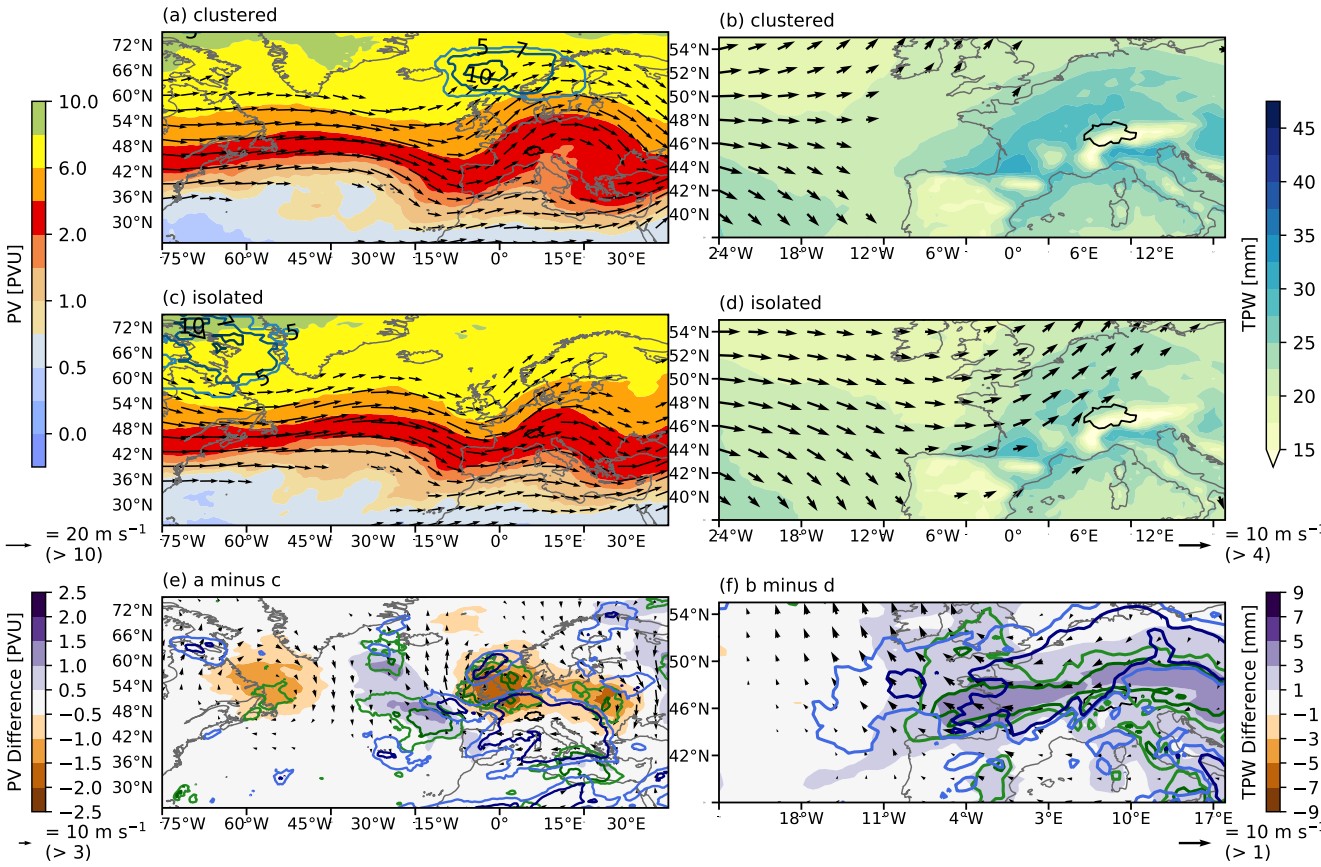

**Figure 5.** For hail events north of the Alps, left column: Potential vorticity at 335 K (color shading; in PVU), wind at 250 hPa (vectors, in m s$^{-1}$) and atmospheric blocking frequency (dark green contours for 5 %, 7%, and 10%). a) clustered hail days (n = 135 days), b) isolated hail days (n = 69 days), e) difference a minus c of PV at 335 K (color shading) and winds (vectors). Statistically significant differences for more than 50% (lighter contour lines) and more than 80% of the resample composites (darker contour lines) of all 500 ks-test and FDR-corrected p-values are shown in e) in green for PV at 335 K and in blue for wind at 250 hPa. Right column: TPW (filled contours, in mm) and wind at 850 hPa (vectors). b) clustered hail days (n = 135 days), d) isolated hail days (n = 69 days), f) difference b minus d. Using the same framework as in e), significant differences are shown in green for TPW and in blue for wind at 850 hPa. The brackets below the wind vector legends indicate the minimum wind speed that is visualized. The grey contours show coastlines, and the black contour line shows the border of Switzerland.



**Figure 6.** For hail events north of the Alps, left column: a) and c) daily maximum T2M (color shading; in °C) and daily mean sea- level pressure (black contour lines, labels indicate by how much the MSLP exceeds 1000 hPa in hPa) a) clustered hail days (n = 135 days), c) isolated hail days (n = 69 days). e) Difference a minus c. Statistically significant differences for more than 50 % of the resample composites (lighter contour lines) and more than 80 % (darker contour lines) of all 500 ks-test and FDR-corrected p-values are shown in green for T2M and in blue for MSLP in e) Right column: CAPE (filled contours, J kg$^{-1}$) and front frequencies (red, labelled lines) for b) clustered hail days, d) isolated hail days, f) difference b-d, the areas with > 50 % significant differences in CAPE are shown in blue hashes (> 80 % almost never present).





### 4.3.2 Atmospheric conditions over Europe and the North Atlantic prior to hail events north of the Alps

Composites of the days preceding the hail days illustrate the evolution of the upper-level changes that result in the more meridionally amplified flow over Europe on clustered hail days. These composites each consist of 31 days (see Table 1). Three days prior to the clustered hail events north of the Alps, the trough over western Europe at 18°W exists, and a ridge is present over central Europe (Fig. 7a). On at least 10% of the days, an atmospheric block is present over Scandinavia north of that ridge. The flow is highly diffluent upstream of the ridge, and the zonal flow over Europe is very weak. This diffluence sustains the meridional amplification of the upstream trough. One day later, the upstream trough over the east Atlantic widens zonally (Fig. 7d). The ridge over Central Europe starts tilting anticyclonically at d-2 and so does the downstream trough (Fig. 7d and g).

The troughs and ridges over Europe strongly amplify from d-3 to d-1 before isolated hail days (Fig. 7b, e and h). Over the North Atlantic a ridge amplifies at 40°W, and atmospheric blocking is present over the western and central Atlantic (Fig. 7b, e and h). The moisture content of the atmosphere also increases by 2mm from d-2 to d-1 (Fig. 8b, e and h).

In summary, the flow is more diffluent and meridionally amplified over Europe three days prior to clustered hail days compared to isolated hail days. Prior to both clustered and isolated hail days, the local atmospheric moisture content increases slightly from d-2 to d-1.



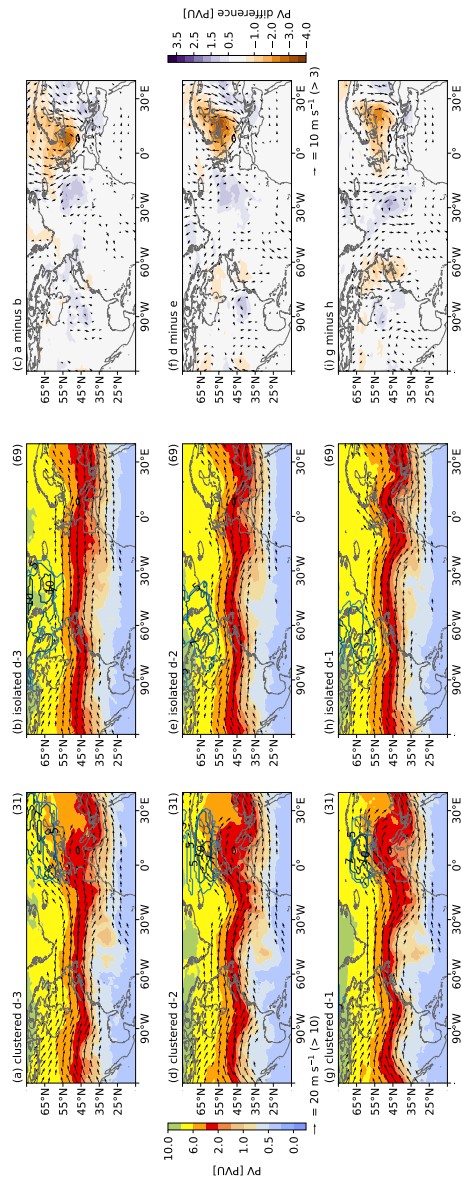

**Figure 7.** PV at 335 K (colored contours), wind at 250 hPa (vectors) and blocking frequency (contour lines) for the three days prior to clustered (left column) and isolated (central column) hail days north of the Alps. The differences are shown in the right column. See caption of Fig. 5 for details.

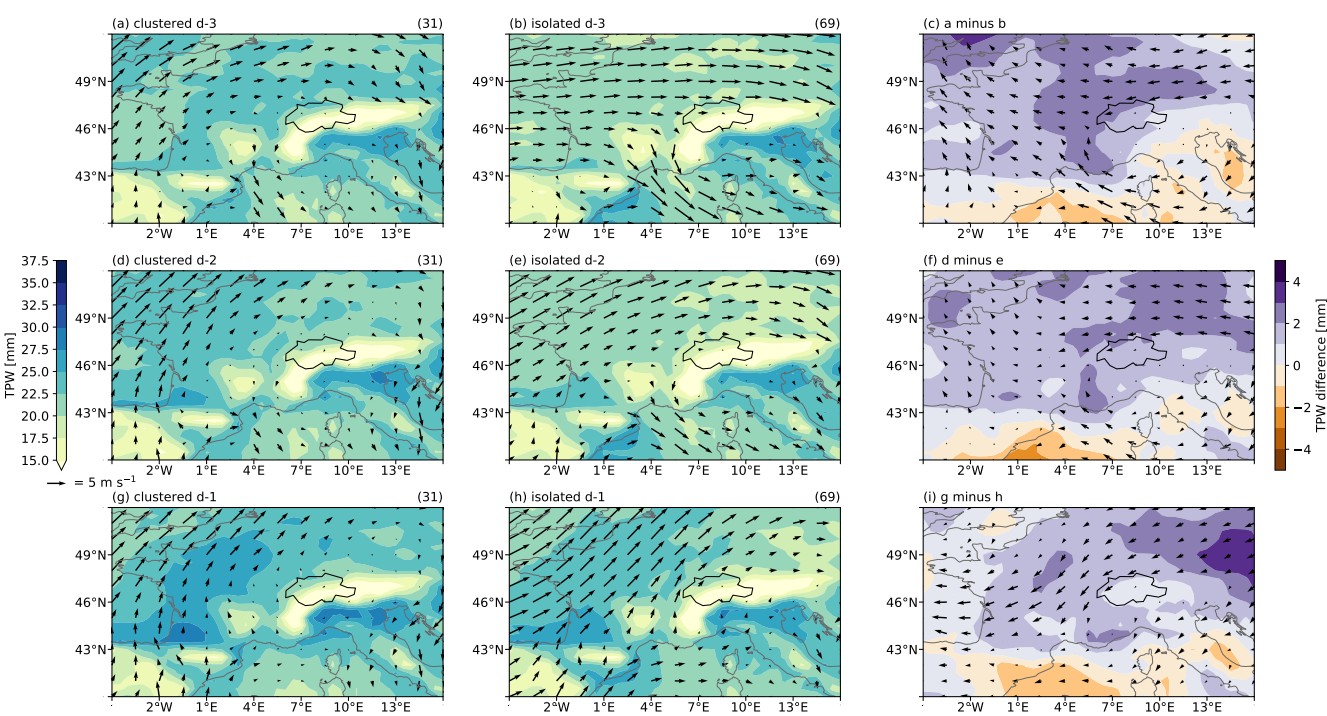

**Figure 8.** TPW (colored contours) and wind at 850 hPa (vectors) for the three days prior to clustered (left column) and (central column) isolated hail days north of the Alps. The differences are shown in the right column. See caption of Fig. 5 for details.



### 4.3.3 Atmospheric conditions over Europe and the North Atlantic on hail days south of the Alps

Similar to the north side of the Alps, a trough over western Europe is located further west on clustered days ( 2°W) (Fig. 9a) than on isolated days ( 5°E) (Fig. 9c). The trough in the clustered days composite is tilted anticyclonically in the subtropics. The downstream ridge over Europe is centered at  18°E in both composites, pointing to a longer wavelength and hence slower propagation of the waves on the clustered days. The ridge over central Europe is more amplified in the clustered composite (Fig. 9e). The downstream trough over the Black sea and the eastern Mediterranean tilts anticyclonically in the clustered days 235 composites (Fig. 9a). No blocks are present over Europe during either clustered or isolated hail days.

In the lower troposphere, the winds on clustered hail days are on average weak ($< 4$ m s$^{-1}$) in and around Switzerland (Fig. 9b). On isolated hail days, westerly winds are slightly but significantly stronger north of the Alpine ridge (Fig. 9d). The moisture content of the atmosphere is higher over most of Europe during clustered hail days (Fig. 9b and d) and marginally 240 significantly higher (1-3 mm) over southern Switzerland (Fig. 9f).

Daily maximum temperatures during clustered hail events are significantly warmer by 2–3K south of the Alps and by 1–2K over the Mediterranean Sea close to Italy (Fig. 10a, c, and e). The differences in mean sea level pressure between clustered and isolated hail days south of the Alps show a weaker low-pressure area during clustered days east of Denmark and south of the 245 Alps (marginally significant, Fig. 10e), weaker high-pressure area over the east Atlantic (not significant), and hence a weaker north-south pressure gradient upstream of the Alps. The mean sea level pressure is higher north of the Alps than south of the Alps. Mean sea level pressure is significantly higher on clustered hail days in a band along the southern edge of the Alps (Fig. 10e), and there is hence a stronger pressure gradient across the Alps on isolated hail days. On clustered hail days, cold fronts are more often present northwest of the Alps (Fig. 10b) and over the Bay of Biscay than on isolated hail days (Fig. 10d). In 250 contrast, more cold fronts are located directly over the Alpine ridge on isolated hail days than on clustered hail days.

CAPE values are statistically significantly larger on clustered hail days over parts of Italy (Fig. 10f), the northern Adriatic (difference $>800$ J kg$^{-1}$), and the Gulf of Genoa. In the study region south of the Alps, CAPE values are insignificantly larger by 350-400 J kg$^{-1}$ on clustered hail days than on isolated days. No significant differences in the wind shear are found over 255 Switzerland (not shown).



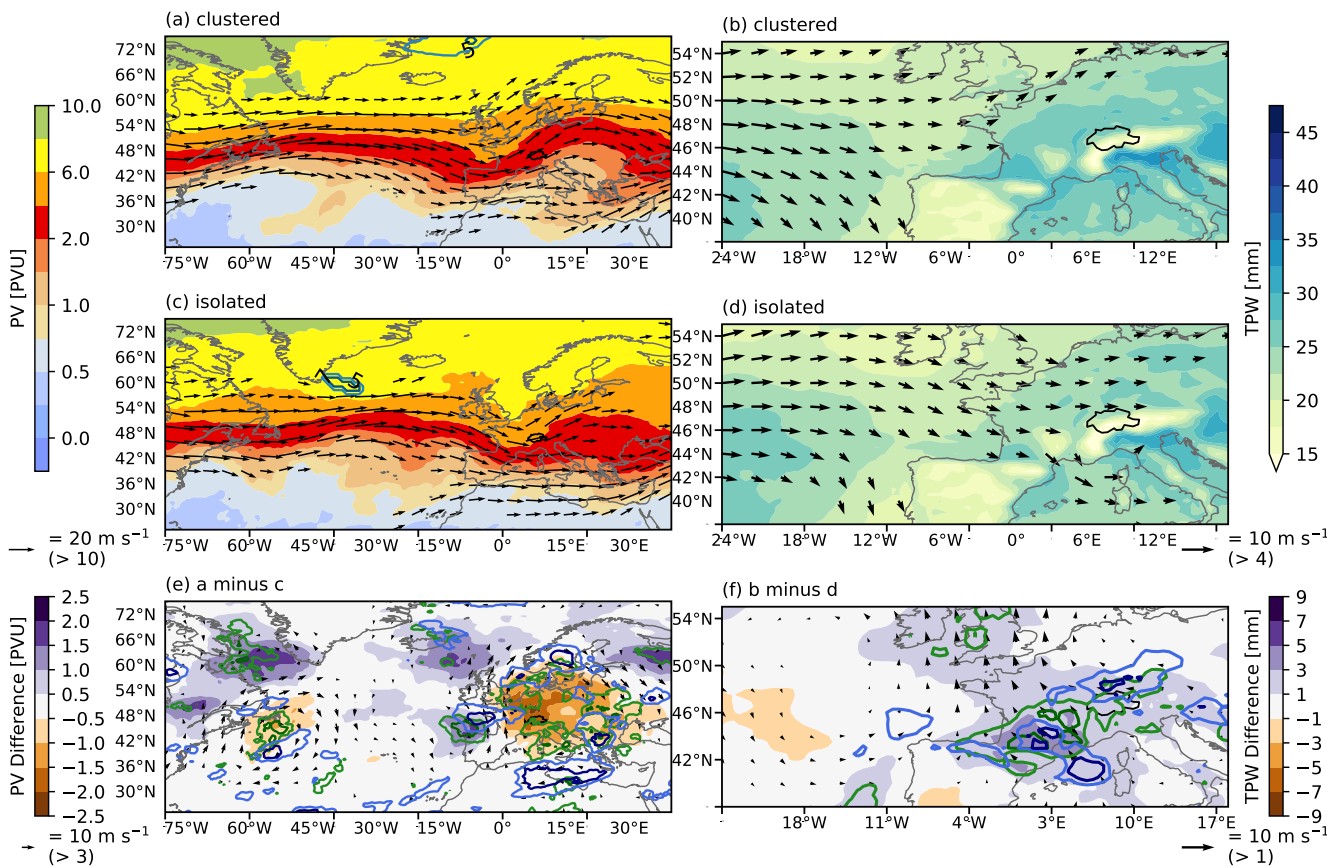

**Figure 9.** During clustered and isolated hail days south of the Alps: left column: PV, winds at 250hPa and atmospheric blocking, right column: TPW and winds at 850hPa. In e), significant differences in 250hPa winds are shown with blue contours and in PV with green contours. In f), the significant differences in 850hPa winds are shown with blue contours and in TPW with green contours. See Fig. 5 for more details.

**Figure 10.** During clustered and isolated hail days south of the Alps: left column: T2M and MSLP, right column: Fronts and CAPE. In e) the significances of differences for MSLP (T2M) are shown in blue (green) and in f) the significances of differences for CAPE are shown in blue hashes. See Fig. 6 for more details.





### 4.3.4 Atmospheric conditions over Europe and the North Atlantic prior to hail events south of the Alps

The Rossby wave pattern over Europe during clustered hail days exhibits a stronger ridge over central Europe compared to the pattern on isolated hail days. Composites of the days preceding the hail days illustrate the evolution of the flow resulting in this
ridge formation.

Three days prior to clustered hail events, a trough is present at 18°W and a ridge upstream is centered at 50°W (Fig. 11a). The trough at 18°W is tilted cyclonically. Although atmospheric blocking is present over the northeastern Atlantic and Scandinavia on d-3, the blocked area decreases in the following two days (Fig. 11a, d, and g). Over western Europe, the flow is
southwesterly, and a small ridge is present at d-3. A downstream trough is present over Greece. Both the ridge over the central North Atlantic and the trough at 18°W amplify over the next two days, and a strong southwesterly flow remains present over western Europe. Over the same period, the ridge over central Europe amplifies, and the downstream trough over Greece breaks anticyclonically. Hence, a typical example of downstream wave propagation is visible in the lagged composites.

The moisture content of the atmosphere over the study region south of the Alps increases by 3 mm between d-3 and d-2 in the composite of the clustered hail days (Fig. 12a and d). Over the study region, the air contains 1–3 mm more TPW prior to clustered days than to isolated hail days (Fig. 12c).

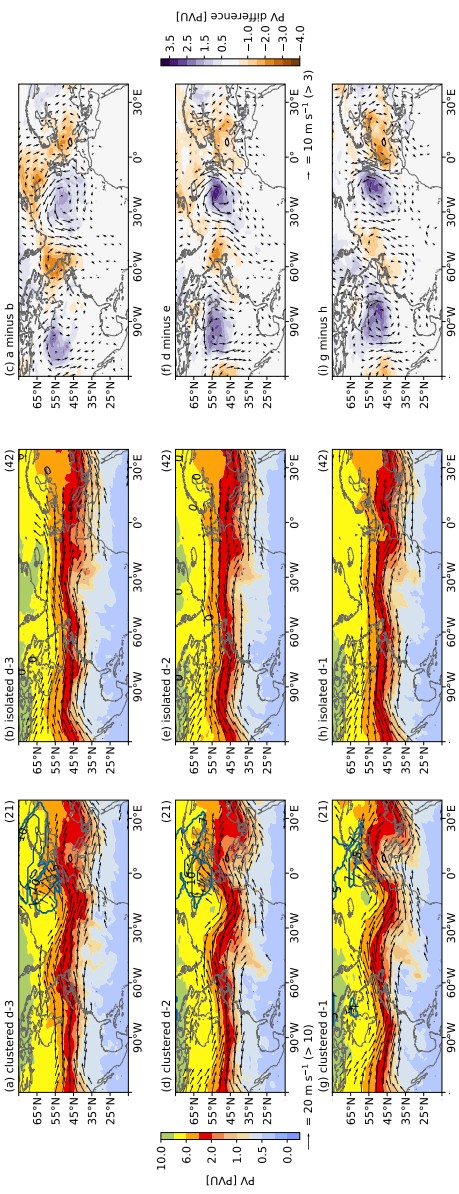

**Figure 11.** Composites of PV at 335 K, winds at 250 hPa, and atmospheric blocking frequency for the three days prior to clustered (left column) and isolated (central column) hail days south of the Alps. The differences are shown in the right column. See caption of Fig. 5 for details.

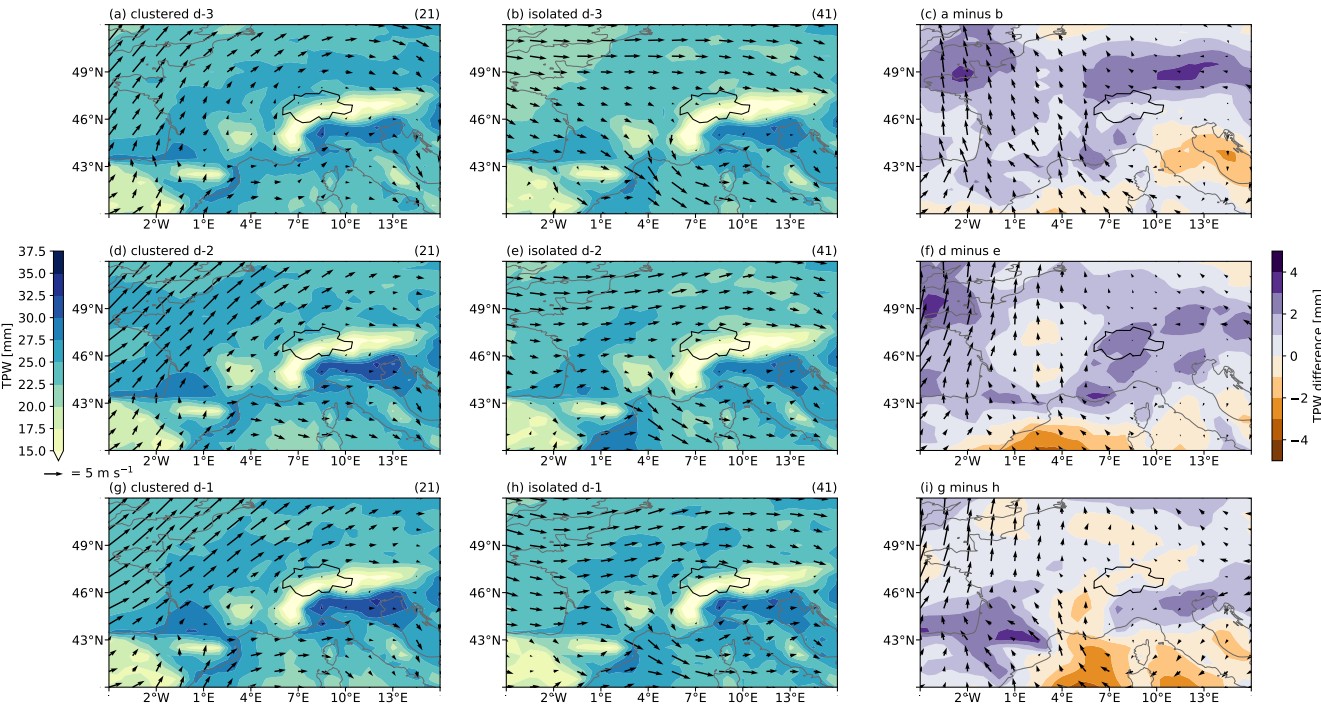

**Figure 12.** Composites of TPW (colored contours) and winds at 850hPa (vectors) for the three days prior to clustered (left column) and isolated (central column) hail days south of the Alps. The differences are shown in the right column. See caption of Fig. 5 for details.





### 4.3.5 Concurrent clustered hail days north and south of the Alps versus hail days that are clustered only south of the Alps

During clustered hail days only south of the Alps, the dynamical tropopause (2 PVU contour) is located over southern France. When clustered hail days also occur north of the Alps, the dynamical tropopause is located further north (Fig. 13a, c, and e). During clustered hail days on both sides of the Alps, fronts are frequent northwest of the Alps and over the Bay of Biscay. The frontal frequencies on clustered hail days only south of the Alps are very different. The frequency of fronts is highest on top of the Alpine ridge and close to zero northwest of the Alps (Fig. 13b, d, and f). The daily maximum CAPE values are

higher ($> 800$ J kg$^{-1}$) over most of the Mediterranean, the difference culminating at $>1200$ J kg$^{-1}$ in the gulf of Genoa and west of Corsica and Sardinia (Fig. 13f). When clustered hail days occur only south of the Alps, thunderstorms seem to be influenced more by convergence zones with stronger southwesterly low-level winds and nearby fronts. During clustered hail days north and south of the Alps, fronts are located further away to the northwest of the Alps, and the local instability governs thunderstorm activity south of the Alps with high values of CAPE.



**Figure 13.** PV at 335 K, horizontal winds at 250 hPa (left column), CAPE and fronts (right column) during clustered hail days occurring only south of the Alps (top row), during clustered hail days occurring both north and south of the Alps (middle row), and their differences (bottom row; significances not shown). See Fig. 5 and 6 for details.





## 5    Summary and discussion

The large-scale and local-scale atmospheric conditions during and prior to multi-day hail clusters and isolated hail days in northern and southern Switzerland are characterized and compared. Hail days between April and September 2002–2019 are defined for a region within the range of Swiss radar stations north of the Alps and for a similar region south of the Alps (Fig. 1). The atmospheric situations are described using a weather type classification and large-scale and local-scale atmospheric conditions.

Clustered hail days occur only during the summer months, whereas isolated hail days also occur earlier and later in the year (Fig. 2 and 3). Differences between isolated and clustered hail days in the prevailing central European weather types are small (Fig. 4).

### 5.1    Atmospheric conditions prior to and during hail events north of the Alps

Several characteristics of the large-scale flow over the North Atlantic and Europe point to more stationary flow conditions on clustered hail days than on isolated hail days (Fig. 5). The flow is more amplified meridionally, and Rossby waves break down-stream of Switzerland, resulting in weak upper-level zonal winds over central and eastern Europe and weaker surface zonal winds. In addition, blocking anticyclones over Scandinavia can contribute to more persistent flow over central Europe (Mohr et al., 2020). There are fewer fronts over western Europe on clustered hail days than on isolated hail days, and the fronts are located further from northern Switzerland (Fig. 6). This difference in front locations points to thermotopographic winds being more relevant for convection initiation during clustered hail days (see e.g., Trefalt et al., 2018; Schemm et al., 2016), whereas prefrontal convergence and prefrontal orographic flow could be more relevant for the initiation of hailstorms on isolated hail days (Schemm et al., 2016; Nisi et al., 2020). On clustered hail days, the air is more humid and warmer by $> 2$–3 K in Central Europe north of the Alps, and CAPE is on average 400-800 J kg$^{-1}$ higher than on isolated hail days over a large area north of the Alps. Hence, instability is substantially higher on clustered hail days, and the stationary flow allows these conditions to persist. The slow-moving large-scale flow signal suggests that clustered hail days might be more predictable than isolated hail days (e.g., Trapp, 2014; Dalcher and Kalnay, 1987).

A high atmospheric moisture content across western and central Europe may be needed for sustained convection over several days north of the Alps. Because northern Switzerland is about 600 km away from oceanic moisture sources, evapotranspiration over land is an important moisture source in summer (Sodemann and Zubler, 2010).

The higher temperatures, higher CAPE values, and the location of blocks during clustered hail days compared to isolated hail days agree well with the differences in local conditions between a month with many hail days and a month with few hail days in northern Switzerland described in Madonna et al. (2018).



Upper-level Rossby waves over the Atlantic are more amplified meridionally prior to clustered days than isolated days (Fig. 7). In addition, blocking over Scandinavia on days prior to clustered hail days may contribute to a diffluent flow over Europe that amplifies the troughs that reach Europe from upstream (e.g., Shutts, 1983). The local moisture content increases by 3 mm prior to both clustered and isolated hail days.

### 5.2 Atmospheric conditions prior to and during hail events south of the Alps

Differences in the large-scale flow conditions between clustered and isolated hail days south of the Alps are similar to the differences north of the Alps with the distinction that atmospheric blocks do not occur over Europe. The large-scale flow is more stationary during clustered hail days than isolated hail days. Locally, the atmosphere is slightly warmer and contains more humidity on clustered days; however, the difference in local CAPE between isolated and clustered days is small and not significant (Fig. 9 and 10). The pressure gradient across the Alps is stronger on isolated hail days, and slightly more fronts are present south of the Alps on isolated hail days. Hence, the favorable conditions for hail are more transient on isolated hail days due to a less stationary large-scale flow, and the stronger cross-Alpine pressure gradients may indicate that Foehn winds support short-lived prefrontal convergence zones. The trough just west of Switzerland and the fronts located directly over the south side of the Alps on isolated hail days typically produce a low-level convergence with thermal winds over this area. These conditions are known to last only several hours and to develop severe hailstorms in the southern Prealps (Luca Nisi, personal communication).

Prior to clustered hail days south of the Alps, the Rossby waves over western Europe have a larger meridional amplitude than prior to isolated hail days (Fig. 11). A trajectory analysis could assess whether this strong amplification results in the transport of moist air masses from the subtropics towards the Alps.

South of the Alps, 58 % of isolated hail days are outside of the seasonal window within which clustered hail days occur. This may be related to the different convective environments of hailstorms in the middle of the convective season and in the shoulder seasons. In midsummer, the supply of humid and unstable air from the Mediterranean towards the Alps ahead of the trough over the eastern Atlantic may create conditions favorable for hail day clustering. The unstable air masses do not require additional synoptic forcing for the formation of hailstorms. Given the slow propagation of the trough eastward, additional moisture and warm air can be advected towards the Alps on subsequent days. Furthermore, strong radiative heating over the Alps, the resulting thermo-topographic flows, and the evapotranspiration of moisture may support hail day clustering. In contrast, at the beginning and end of the hail season, the air masses are not as unstable to begin with and therefore need stronger triggers to produce hailstorms.

Even though the flow over the Atlantic on isolated hail days resembles the flow during the negative North Atlantic Oscillation (NAO) phase, we do not find any significant correlation between the NAO index and the isolated hail days (not shown). This is in agreement with Piper and Kunz (2017), who comment that convective activity in southern Switzerland can occur regardless of large-scale forcing thanks to the complex orographic mechanisms. More fronts occur over the Alpine ridge during isolated





hail days and northwest of Switzerland during clustered hail days.

Different atmospheric conditions are associated with clustered hail days on both sides of the Alps than with clustering only south of the Alps (cf. Fig. 13). When clustered hail days occur only south of the Alps, stronger winds and bulk wind shears combined with lower CAPE values suggest a stronger dynamic forcing of the thunderstorms. On clustered hail days affecting all of Switzerland, CAPE values are $> 1200$ J kg$^{-1}$ larger over the Ligurian sea, and the winds and bulk wind shear are weaker. Furthermore, fronts almost never occur directly over the Alpine ridge and are mostly located over the Bay of Biscay or 400 km northwest of the Alps. In this situation, both prefrontal convergence zones and orographic heating may well be the more

likely drivers of convective activity.

## 6 Conclusions and outlook

Multi-day hail clusters are a regular phenomenon in Switzerland both north and south of the Alps. We observe on average 10 clustered days per year in the north and 6 clustered days in the south. We compared the large- and local-scale conditions prior to and during multi-day hail clusters and isolated hail days between 2002 and 2019, within the range of Swiss radar stations.

Multi-day hail clusters occur only between mid-May and end of August on the north side of the Alps and between mid-June and mid-August on the south side of the Alps, whereas isolated hail days occur during the entire convective season.

For the regions both north and south of the Alps, the large-scale atmospheric flow over the east Atlantic and Europe prior to and during clustered hail days is more amplified meridionally and characterized by a trough located in the east Atlantic.

The meridional amplification is enhanced by atmospheric blocks located over Scandinavia prior to clustered hail days. On the north side of the Alps, furthermore, warmer and more humid local conditions with significantly higher CAPE values are found during clustered hail days. Fronts northwest of Switzerland are located farther away than on isolated hail days. Our findings suggest that on the north side of the Alps, thermotopographic winds are more relevant for convection initiation during clustered hail days, whereas prefrontal convergence and prefrontal orographic flow may be more relevant for the initiation of hailstorms

on isolated hail days.

The local conditions south of the Alps are warmer and more humid on clustered hail days; however, differences in local CAPE between clustered and isolated hail days are not significant. We observe a stronger pressure gradient across the Alps on isolated hail days, which may indicate that Foehn winds support short-lived prefrontal convergence zones south of the Alps.

On isolated hail days south of the Alps, local conditions supporting convection are dispersed faster by the large-scale flow.

During hail days that are clustered both north and south of the Alps, fronts are frequent over the Bay of Biscay and 400 km northwest of Switzerland. In contrast, on hail days that are clustered solely south of the Alps, fronts are almost exclusively located over the Alpine ridge. Furthermore, low-level winds are stronger south of the Alpine ridge, and CAPE values are lower





north of the Alps and over the Gulf of Genoa. This suggests that when hail days cluster only south of the Alps, dynamic pro-
cesses are responsible for maintaining convective conditions over several days.

Future research could compare the characteristics of hailstorms between clustered and isolated hail days, such as their
duration, speed, and direction of movement, and the hour of the day at which they are most likely to occur in which regions of
Switzerland. These characteristics could provide more insight into the likely trigger mechanisms during and isolated hail days.
The results of this study also pose a question: Do average higher daily maximum temperatures, the weaker zonal flow, and the
meridionally amplified atmospheric waves on clustered hail days mean that climate warming may increase the frequency of
multi-day hail clusters?"

**Appendix A:   Selecting the area threshold to define hail days**

The method used to choose the area threshold is shown in Table 2. The number of days on which grid- boxes with POH values
$\geq 80\%$ cover at least $200\,\mathrm{km}^2$ are counted for the region north of the Alps (top rows) and the region south of the Alps (bottom
rows). Different percentiles (p) of all nonzero POH values $\geq 80\%$ areas are calculated for each region (area in $\mathrm{km}^2$). The
days are divided into whether their area value is greater than the percentile (area $\geq$ p) or not (area $<$ p) and whether the days
observed at least five car insurance loss reports ($\geq 5$ losses) or not ($< 5$ losses). In an ideal case, the columns "$\geq 5$ losses"-
"area $\geq$ p" and "$< 5$ losses"- "area $<$ p" would have very large numbers of days, and the other two columns would have very
low numbers. North of the Alps, 285 days have an area $\geq$ 80th percentile and 192 days an area $<$ 80th percentile. Of the 285
days, 167 have at least 5 losses, more than the 118 days with less than 5 losses. Furthermore, the 192 days include only 23 days
with $\geq 5$ losses, compared to 96 days with $< 5$ losses. South of the Alps, the 80th percentile also provides the most balanced
numbers of days per category while guaranteeing that more days are defined as "hail days" than "not hail days".

**Appendix B:   Methods: Details on resampling considering the seasonality of clustered hail days**

Here we describe the methods for determining which hail days we count as within independent clustered hail day periods. From
these, we create the average composites of reanalysis variables during clustered hail days. We then explain how the isolated
hail days are resampled following the seasonality of clustered hail days. Furthermore, details of the Kolmogorov-Smirnov (KS)
test and the modification of the significance threshold through the false discovery rate (FDR) are explained.


The clustered hail days are by nature dependent. We therefore apply a 500-times-repeated resampling to all clustered and
isolated hail days such that each of the 500 series contains only serially independent data. Isolated hail days are by nature
independent; this category does not need any additional treatment to ensure independence. However, clusters of hail days that
are longer than 11 days are further divided into periods of 5 days that have at least 2 days between each other. We call these
periods independent. For the clustering period in 2004 (Fig. 2), some clustered hail days have the sequence no hail (0) and





**Table A1.** Number of days >200 km$^2$ per area percentile (p) and number of car insurance loss reports (losses).

| | percentiles (p) | Area [km$^2$] | >5 losses | <5 losses | # hail days | >5 losses | <5 losses | # nonhail days |
|---|---|---|---|---|---|---|---|---|
| | | | area >p | | | area <p | | |
| **North of the Alps** | 70 | 164 | 190 | 224 | 414 | 0 | 0 | 0 |
| | 75 | 319 | 184 | 184 | 368 | 6 | 40 | 80 |
| | **80** | **580** | **167** | **118** | **285** | **23** | **96** | **192** |
| | 85 | 990 | 146 | 72 | 218 | 44 | 152 | 304 |
| | 90 | 1881 | 120 | 37 | 157 | 70 | 187 | 374 |
| **South of the Alps** | 70 | 188 | 119 | 203 | 322 | 0 | 0 | 0 |
| | 75 | 316 | 108 | 164 | 272 | 11 | 39 | 78 |
| | **80** | **499** | **94** | **123** | **217** | **25** | **80** | **160** |
| | 85 | 755 | 76 | 90 | 166 | 43 | 113 | 226 |
| | 90 | 1330 | 55 | 48 | 103 | 64 | 115 | 230 |

hail days (1) "11011101". Although all these hail days are by our definition clustered, the central 5-day period that starts and ends with no hail days "01110" contains only 3 hail days, despite being marked as clustered by their attribution to neighboring 5-day periods. If in such cases the algorithm determining independent periods by accident selects a sequence containing only 3 hail days, that choice is corrected by displacing the 5-day period to one day earlier. Consequently, the number of hail days per

clustering period is always ≥4. This criterion of independence has the consequence of not including all potentially available clustered hail days. North and south of the Alps, this treatment additionally removes 29 and 13 out of 164 and 102 clustered hail days, respectively. The resampling vectors north of the Alps each have 32 days and south 21, following the number of independent clustering periods in each study area. The seasonality following which isolated hail days are sampled is defined as follows. The number of clustered hail days within independently clustered hail day periods is counted per 20-day period

as shown in Fig. 3, starting at DOY 80–99 and ending at DOY 260–279. Wherever clustered hail days occur, the relative frequencies of clustered hail days per 20-day period are divided by the relative frequencies of isolated hail days. These values are the probability of sampling per DOY of all isolated hail day during each 20-day period. Because the number of isolated hail days per DOY varies, the probability of sampling per each isolated hail day is further divided by the number of isolated hail days per DOY. The two-sample KS-test measures the largest distance between the two empirical cumulative distribution

functions) of both data samples. It has the advantage of being independent of the distribution of each individual data set. Hence, for each variable, the KS-test is applied 500 times, comparing each resampled series of clustered hail days to its isolated-hail-



day counterpart (first resampled vector of clustered hail days with first resampled vector of isolated hail days; second with second, third with third, etc.). For atmospheric fields, this procedure yields 500 ks-values and 500 $p$-values for each grid-point. The $p$-value indicates how likely it is that the two compared vectors stem from the same distribution (null hypothesis) and

that the difference is statistically insignificant. Statistical tests typically consider a significance level $\alpha$ of 5%. Following this method, the null hypothesis is rejected if the chance of accepting it is less than 5%. However, with $N$ repeated tests and an $\alpha$ of 5%, on average $N*0.05$ (here $500*0.05 = 25$) test results will falsely reject the null hypothesis (Type I error). That number itself is drawn from a probability distribution whose mean is $N*0.05$ and can vary considerably with an increasing $N$. A solution is to control the false discovery rate (FDR, Wilks, 2016). The FDR-corrected threshold does not define the probability

of falsely accepting the null hypothesis ($p$-value) but the probability that a rejected null hypothesis should actually have been accepted ($q$-value). In concrete, we follow the procedure explained in Wilks (2016). The $p$-values are sorted in ascending order and each individual $p$-value $p_i$ is compared to a threshold $p^*_{FDR}$ that varies according to $q$ (in Wilks (2016), $q$ is called $\alpha_{FDR}$), $N$ and $i$. Assuming statistical independence of each of the $N$ local tests $p^*_{FDR}$ is the largest $p_i$ that is equal or smaller than $(i/N)\alpha_{FDR}$:

$$p^*_{FDR} = [p_i : p_i \leq (i/N)\alpha_{FDR}] \tag{B1}$$

*Data availability.* The ERA5 and ERA-Interim reanalyses used in this study can be accessed from the ECMWF website (https://www.ecmwf.int/en/forecasts/datasets/reanalysis-datasets/era5 and https://www.ecmwf.int/en/forecasts/datasets/reanalysis-datasets/era-interim, last access: 8 April 2021) (ECMWF, 2021a, b).

*Author contributions.* OM and UG developed the initial project idea; HB conducted the main analyses, created Fig. 2–10, and wrote the

paper together with OM, KS and UG; KS provided the radar data and Fig. 1. HB, OM, LN, and KS discussed the results in depth. All coauthors commented and contributed to the improvement of the manuscript.

*Competing interests.* The authors declare that they have no conflict of interests.

*Acknowledgements.* Many thanks go to Sebastian Schemm and Michael Sprenger for kindly providing the ERA-Interim front data set. Furthermore, many thanks go to Andrey Martynov for maintaining the servers and data sets with which the analyses were conducted. Finally,

we thank Simon Milligan for editing this manuscript for its language.

*Financial support.* This project was funded by the Swiss Mobiliar insurance company through the Mobiliar Lab for Natural Risks of the University of Bern.




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
