# Peer review of "Multi-day hail clusters and isolated hail days in Switzerland – large-scale flow conditions and precursors"

_Weather and Climate Dynamics, 2021_

## Referee Comment (RC3)

Review of: "Multi-day hail clusters and isolated hail days in Switzerland – large-scale flow conditions and precursors", by Barras et al.

**Summary**

This study documents and compares the frequency of multi-day hail clusters versus isolated hail days in Switzerland. It shows that multi-day clusters are relatively common during summer, and that conditions relating to large, longwave troughs favor their occurrence.

I think this is solid research, and a well-designed and well-written paper. My comments are relatively minor, and mostly regard clarifications.

**Comments**

Figure 1: some lat/lon information on this figure would be useful (it's not clear to me what the km values relate to

line 92: I assume that such a calculation of BWS is standard practice for this part of the world? Because of the complex topography, does using the 10 m winds i/o 850 hPa lead to excessively noisy BWS analyses?

line 105: Just to confirm that I understand this: the percentiles here are based on the set of daily areas over which POH>80%?

line 108: This is a bit confusing/ I assume that 'nonhail days' here refers to days *not* identified by the POH footprint criteria? (rather than an "actual" nonhail day?) I'm trying to reconcile >5 car insurance hail losses in absence of hail.

line 118: While I appreciate the value in following the published approach of other researchers, I think it's worth giving a very brief justification (meteorological or otherwise) on the significance of 5-day periods. To be clear, I think I can infer the justification, and I don't have an issue with this period of consideration, but it is fair to ask why 5 is preferable.

line 150: s/b 'resampled'

line 363: Using your analysis, you could also express this in terms of a probability: Given a hail day, there is an xx % probability that this day is part of a multi-day cluster.

line 388 (future research): One of the cited North American studies speculated that upscale feedbacks driven by the diabatic heating of deep convection may contribute to the consecutive days of hazardous convection. Not knowing the total size and duration of the convective storms contributing to clustering, it's difficult to know whether this idea has any relevance to Switzerland, but it's perhaps something to consider. If relevant,

this would have implications, for example, on the necessary approaches to climate modeling (resolution, global versus limited-area, convective parameterizations, etc.).

---

## Author Response (AR1)

**Answers to peer reviews of wcd-2021-25**

We would like to thank the three reviewers, Pieter Groenemeijer, Harald Richter, and the third anonymous reviewer for taking the time to read and review this article. Their comments, questions and suggestions are of great value and contributed to improving this article very positively.

The answers have been added to the submitted reviews in golden brown and new/altered parts of the article are in blue. Line numbers written in golden brown indicate the line numbers of the newly submitted article and those in red and brackets are the same line numbers in the track-change version of the article.

**1st review**

**Comment on wcd-2021-25**

Pieter Groenemeijer (Referee)

Referee comment on "Multi-day hail clusters and isolated hail days in Switzerland – large-scale flow conditions and precursors" by Hélène Barras et al., Weather Clim. Dynam. Discuss., https://doi.org/10.5194/wcd-2021-25-RC1, 2021

General comments:

The authors present a clear analysis of the differences between the synoptic-scale conditions associated with clustered and isolated hail events in Switzerland, both north and south of the Alps, and discuss the differences between those conditions. The steps in their analysis are described clearly and the figures illustrating the analysis are of high quality. In my opinion, the only point that could be improved a little is to elaborate a bit on why contrasting isolated from clustered events is so important. Overall, the paper is of high quality and I recommend it for publication with only minor revisions.

Thank you very much.

Specific comments:

Lines 30-34:

*"For insurance companies, an important question is whether hail events can be considered as independent or not. "*

I suggest explaining this a bit more precisely. What do insurance companies mean by independent, and why is this important to them? Besides this point, are there other reasons why clustered events are interesting? Is the hail usually larger or more widespread during those events?

Thank you for the questions. For insurance and reinsurance companies in Switzerland, it is of interest to know whether two or more damages that occurred within a short time period

(few to several days) have the same tectonic or atmospheric cause or not (see Swiss Supervisory Regulation (AVO), Art. 176[4]). If they do, then the client (individual to insurance or insurance to reinsurance) will have to pay the deductible only once. If the two events are considered to be related to different tectonic or atmospheric events, then the client will have to pay the deductible twice or more. Delineating an atmospheric event is, however, not straightforward. We conducted this research to find out if and which different atmospheric conditions define and lead to isolated hail days versus clustered hail days.

Being able to predict clustered hail events may also be of interest in case they are concurrent with other natural hazards, such as heavy precipitations. Combined, they may cause other or more damages and dangers than apart.

About the size of hail on clustered vs. isolated hail days: Next to POH, MeteoSwiss also operationally calculates the radar- and model-based hail algorithm maximum expected severe hail size (MESHS; Joe et al. 2004, based on Treloar, 1998). It estimates the maximum diameter ≥ 2 cm that is expected in a square kilometer and is calculated similarly to POH. So far, we had not looked at the differences in POH or MESHS area sizes of clustered and isolated hail days. It seems, however, that your suspicion is correct. The area sizes for all three criteria POH ≥ 80 %, MESHS ≥ 2 cm and MESHS ≥ 4 cm are significantly larger during clustered than isolated hail days north of the Alps (see Figure R1). South of the Alps, the area sizes are larger for all three criteria as well, but not significant for MESHS ≥ 4 cm. We can, therefore, conclude that hail is usually larger and more widespread during clustered than during isolated hail days. This finding had initially not been a motivation for this research, but we now added it in lines 34-35 (35-36).

Lines 32-35 (32-36) now explain the motivation with more precision:

"For insurance companies, the question is whether losses separated in time and space can be traced to the same atmospheric or tectonic cause (see Art. 176 para. 4, Swiss Insurance Supervision Ordinance, 2006) and, therefore, are part of one event. Furthermore, we find that multi-day hail events are associated with significantly larger areas affected with hail than isolated events (see Fig. A1 in the Appendix)."

[Figure]

Figure R1: Boxplots of daily areas [km$^2$] with POH ≥ 80 %, Maximum expected severe hail size (MESHS) ≥ 2 cm and MESHS ≥ 4 cm for clustered (dark) and isolated (light) hail days north of the Alps (blue) and south of the Alps (green). Note that the y axis is logarithmic. The boxplots are Tukey-style whiskers with notches showing the 95% confidence interval for the median m, given by m ± 1.58*IQR/√n (McGill et al., 1978). IQR is the interquartile range and n is the sample size (see also Krzywinski and Altman, 2014). n is sometimes smaller for MESHS, because on some days POH ≥ 80 % areas exceed the hail-day definition threshold, but MESHS never reaches 2 or 4 cm.

In the new Appendix Figure A1, we only show the boxplots for POH ≥ 80 %, since it conveys the message without MESHS being necessary.

Line 35 onward:

I suggest also citing this relevant publication, in which the synoptic-scale weather pattern and relation to storms is discussed as well:

*van Delden, A., 2001. The synoptic setting of thunderstorms in western Europe. Atmospheric Research, 56(1-4), pp.89-110.*

We added the citation in lines 39-40 (40). This sentence now says:

" Large-scale flow patterns have been linked to the spatial and annual variability of thunderstorms in Europe (van Delden, 2001; Mohr et al., 2019; Piper et al., 2019) and southwestern Europe (Merino et al., 2013), [...] "

Line 86:

*"We extracted ERA-5 variables at a 6-hourly temporal resolution and a spatial resolution of 0.5°. "*

Please indicate as well which vertical resolution you used, so that readers will be able to confirm it was sufficient for calculating Ertel PV.

We added "at all available vertical levels," in the sentence explaining the ERA-5 variable extraction (see line 92 (95)).

Figure 2:

It is not clear to me how there can be events between the first and last day of clustered events, i.e. dots, which are not groups of 2 or 3 days and therefore seem "isolated", that are not classified as such. In other words, why are single dots that are not encircled to indicated an isolated event in the figure? This probably has a very simple answer :-)

We will try to improve the clarity: We consider isolated days only if they are preceded and followed each by 3 non-hail days. There might be single hail days closer to other hail days that are not isolated. The isolated hail day definition has not changed, but the explanation has now been simplified to (see lines 127-128 (130-132)):

" In a period of 5 days we require multi-day clustered hail periods to have at least 4 hail days and isolated hail periods to have only 1 hail day *next to 3 non-hail days, both before and after.* "

Line 345:

*"In contrast, at the beginning and end of the hail season, the air masses are not as unstable to begin with and therefore need stronger triggers to produce hailstorms."*

I don't think triggers need to be stronger to initiate storms when instability is lower. The difficulty here is in the definition of "trigger", which has not been defined in the manuscript. Maybe it is good to be more specific about what you mean by trigger. I consider a trigger to be an area of meso-gamma-scale (2 - 20 km) upward motion sufficiently strong to lift parcels to their level of free convection, which usually also needs to have sufficient longevity and spatial coherence to create a storm. The factors inhibiting initiation despite mesoscale lift are in particular 1) convective inhibition and 2) dry air entraining into the updrafts, not so much the amount of instability as expressed by CAPE. It can be that by trigger, you refer to lift over a much larger scale and for a longer duration, so that it modifies the environment significantly and creates CAPE?

Thank you for the comment. As you mention, the definition of "trigger" needs clarity, but since triggers are not the focus of this paper, we finally preferred to not use the term. Instead, the sentence now says (lines 360-362 (369-371)):

"In contrast, at the beginning and end of the hail season, the air masses are not as unstable to begin with and may therefore need a stronger lifting mechanism to produce hailstorms."

Technical comments:

Affiliation: "Climate Change Reserach". Correct to "Reserach"

Line 18: Duplicate word "Moreover"

Line 200: Remove the excess word "than".

Thank you, these errors were corrected.
* * *
**2nd review**

**Review of WCD-2021-25**

Harald Richter (Referee)

**Title:** Multi-day hail clusters and isolated hail days in Switzerland –large-scale flow conditions and precursors

**Authors:** Hélène Barras, Olivia Martius, Luca Nisi, Katharina Schroeer, Alessandro Hering, and Urs Germann

**Recommendation:** Major revision

**Summary:** The study is examining the synoptic-scale atmospheric patterns during and ahead of series of successive hail days (clusters) and isolated (one-off) hail days over Switzerland. Hail days are characterized based on a radar reflectivity-based proxy (Probability of Hail) in combination with vehicle damage data from an insurance company. The key findings of this ~18-year climatological study is that clusters preferentially occur in more amplified wave patterns with higher atmospheric instability, while isolated hail days occur in a more zonal flow pattern with less instability and stronger dynamical forcing.

**General Comments:** The study supports more rigorously existing notions that amplified and slowly progressing synoptic wave pattern allow for the building of a more significant warm sector and multi-day convection before a frontal passage drives the instability south out of the domain of interest. It combines a diverse range of tools to more objectively identify atmospheric blocks, synoptic patterns and lower- level fronts. The classification into hail days has been thorough and rigorous, with a valuable and often elusive insurance loss dataset strengthening the hail days classification scheme.

The conclusions of the study are not overly controversial, but could be made to be more insightful. In my view there is a missing connection between the larger-scale patterns and the actual storm environments that promote damaging hailstorms. The formation of those hailstorms that results in insurance claims often requires more than atmospheric instability (or some form of CAPE). The growth of larger hail (golf balls and above) may also be associated with dynamically enhanced storms where the vertical shear of the ambient horizontal flow is a key predictor. While the study comments on instability, vertical shear is not derived methodically from the reanalysis data. Of potentially significant advantage would be to reference more local characterizations of the near-storm environment in an often Alpine setting to convincingly collect all those ingredients that subsequently should be extracted from the reanalysis. Have case studies of damaging hailstorms been conducted in the Swiss Alps? What key ingredients have been identified across such studies? Has deep-layer shear been present in the more notable hailstorms, or did damaging hail form through a pulse storm mechanism alone?

On local characterizations of near-storm environments, case studies, key ingredients and on the presence of vertical shear: Yes, the near-storm environments have been studied in Switzerland: Schemm et al. (2016) analyzed the co-occurrence of cold fronts and hail in Switzerland and characterized the 700-850 hPa wind shear during pre-frontal hail formation.

They found that for north of the Alps, BWS has a tendency to increase and reach a maximum 2-4 hours before hail initiation. South of the Alps, this increase is not as pronounced. The BWS therefore may play a role for hail formation when fronts are in the vicinity. Trefalt et al. (2018) conducted a case study of a severe hailstorm that produced damages in the northern Prealps on June 6 2015. The simulated thunderstorms for that day and region formed through convergence at mountain tops (thermo-topographic winds) and the strong directional wind shear that resulted from the large-scale flow and thermo-topographic winds most likely contributed to the longevity of the storm. In June 2021, many parts of western & central Europe were hit by several thunderstorms producing very large hail. In Switzerland, the diameters even reached tennis ball sizes, which has not happened in years if not decades. During these hail days, several supercells occurred, which points to strong wind shear being present. These events do point to bulk wind shear potentially being an important ingredient for very large hail in Switzerland. Trefalt (2017) studied different ingredients during hail days vs. non-hail days and found significant differences in CAPEMU, CINMU, in the 850hPa to 500 hPa lapse rate, the atmospheric moisture content and specific humidity at 2 m above ground. The 0-3, 0-6 and 3-6 km bulk wind shear, however, was not significantly higher during hail days than non-hail days. Looking at the 18 years between 2002-2019, we have found that bulk wind shear does not play a crucial role for the longevity of hail events. We compared the 850-500 hPa bulk wind shear composites between clustered and isolated hail days (see Figures R2 and R3 below) and found no significant differences. For hail events north of the Alps, the closest larger areas with significant differences are in the Po valley, over the Adriatic sea and central Italy.

In the article we mention windshear in the data section:

Lines 99-100 (102-103): "Bulk wind shear values are obtained by subtracting the horizontal wind components at 850 hPa from the wind at 500 hPa."

We also mention wind shear in chapters 4.3.1, 4.3.3 and 5.2:

Line 213 (218) and 265-266 (273-274): "No significant differences in the bulk wind shear are found over Switzerland (not shown)."

Lines 372-375 (381-384): "When clustered hail days occur only south of the Alps, stronger winds and bulk windshear combined with lower MUCAPE values suggest a stronger dynamic forcing of the thunderstorms. On clustered hail days affecting all of Switzerland, MUCAPE values are > 1200 J kg-1 larger over the Ligurian sea, and the winds and bulk wind shear are weaker."
→ In the particular cases of clustered hail days only south of the Alps, wind shear may play an important role, however these cases require further and closer investigation with, as the reviewer mentioned, a less coarse data set.

We now also added another sentence in the discussion section 5.2 in lines 368-369 (377-378):

"Furthermore, composites of bulk wind shear did not show any significant difference in the local environment between clustered and isolated hail days (not shown)."

[Figure]

Figure R2: Bulk wind shear (green contours with labels, in m s$^{-1}$) during a) clustered and b) isolated hail days north of the Alps. c) shows the difference a minus b in orange/blue labeled contours. The blue dashed areas indicate locations in which the differences are significant (50 % level).

[Figure]

Figure R3: Same as Figure R2 but for south of the Alps.

A further note on pulse storms in Switzerland: More notable hail storms in Switzerland (in terms of hail size) are often supercells, especially north of the Alps. Pulsed storms may produce severe hail as well but rather by producing large hail accumulations rather than large hail sizes. The reason may be that irregular updrafts in pulsed storms (high instability but weak wind shear) interfere in building and maintaining hail cores in storms. Nisi et al. (2018) showed that bigger hail sizes are related to longer and long-lasting hail swaths.

A second general point worth addressing is embodied in the significant number of minor comments below. A commonality between a good number of these comments is the language of the manuscript which requires a bit more work to remove pockets of ambiguity in the expressions used (e.g., an easier description of the hail day definition employed), and missing information in a number of places so that a reader can clearly understand the argument presented or simply needs to work less hard to select between possible interpretations or clear up ambiguity. Each individual point is an easy piece of work to clarify, but in aggregate this amounts to a major comment.

In my view, the manuscript is well on its way to publication and would sit nicely in the domain where topography and severe convection interact.

Thank you very much!

**Minor Comments:**

1.  p.1, l.18: The word *moreover* occurs twice. Corrected.

2.  p.2, l.28: second Púcik citation has incorrect format. Corrected.

3.  p.2, par.1: recommend to replace *weather situations* by *weather patterns* or *synoptic patterns.*

    We replaced it with *weather patterns* in the mentioned paragraph and also replaced *weather situation* with *weather pattern* in the title of section 4.3.

4.  p.3, l.64: Waldvogel's dataset was compiled during or before 1979 with a radar or radars now over 40 years old. Is there a risk that the Waldvogel dataset (and therefore the POH polynomial fit) could no longer be representative of today's radar retrievals of the 45 dBZ height due to narrower radar beam widths, improved radar calibration practices, or other factors that have changed over the 40-year period?

    Radar technology evolved substantially over the past decades, but the algorithm developed by Albert Waldvogel for the probability of hail (POH) is relatively simple, only uses reflectivity and is practically not very sensitive to all the technological changes of the past decades. The main prerequisites for POH are a proper calibration and scanning up to high elevation angles for the determination of the echo top. The group of Albert Waldvogel at ETH was using external microwave generators, rain disdrometers and hail spectrometers to rigorously calibrate and monitor the stability of their research radars. The calibration of the ETH research radars operated during dedicated field experiments is comparable to that of today's operational

radars. There may be some differences in the antenna radiation pattern specs, but the impact on the echo top retrieval and POH is marginal for the type of analyses presented in this paper.

5. p. 3, l.64: How is the freezing level obtained for the POH calculations? Is it an operational mesoscale model, or interpolation between sounding sites? Could there be quality issues with such freezing level estimates over complex mountainous terrain?

   The freezing level height is obtained from the numerical weather forecasting model COSMO-1 (see https://www.cosmo-model.org/), which has a spatial resolution of 1x1 km and a temporal resolution of 10 minutes which are aggregated to 1 hour in the model analysis. There are some quality issues, mostly related to inaccurately located simulated thunderstorm cells or small shifts in the timing of frontal passages. These errors have a minimal effect on the daily POH areas and the identification of hail days in our study. The reasons are as follows: For the identification of hail days we first integrate over 24 hours by taking for each pixel the maximum POH, and then integrate in space by counting the number of pixels with daily maximum POH larger than 80%. By integrating in time and space the stochastic errors in POH are smoothed out and one obtains a robust statistic to identify hail days.

6. p.4, l.91: What parcel is being launched for the CAPE calculation – surface, mixed layer, most unstable?

   According to the Copernicus ERA5 data set description page it is the most unstable CAPE (MUCAPE). The description says: "CAPE is calculated by considering parcels of air departing at different model levels below the 350 hPa level. [...] The maximum CAPE produced by the different parcels is the value retained."
   In the entire article, any mention of CAPE has been replaced with MUCAPE.

7. P.4: I am a bit unclear on what is computed off ERA-Interim, and what is computed off ERA-5 (it seems 2 out of 3 schemes run off ERA-I, one off ERA-5, and all stand-alone fields are sourced from ERA-5)

   All reanalysis variables stem from ERA-5, except for the fronts. The fronts were calculated in ERA-Interim.

8. P.5, l.100: Presumably the POH grids are available for every radar scan during a day. Is the daily hail footprint the area (on the 1 m x 1km grid) where POH ≥ 80% for at least one radar scan during that 24-hour period?

   Yes, we look at the area in which the daily maximum POH per grid-box is greater or equal to 80%, considering all radar scans (aggregated to a 1 x 1 km2 resolved grid). Therefore, the overlapping POH ≥ 80% areas of different thunderstorms that occurred at different times during the same 24 hours are counted only once.

9. P.5, first paragraph: given the daily POH footprint as well as the hail day thresholds both use "80", the paragraph confused me at first. Is the 80th percentile taken from

the daily POH distribution (excluding footprints of size zero) for the 10-year period 2003-2012 (i.e., matching the period of the available insurance loss data)?

Yes, we suspected that the 80% and 80th percentile might lead to confusion. We tried to clarify it in the paper with the following change in line 116-118 (120-121):

"As a result, we define a hail day as a day with *a daily maximum POH ≥ 80 % footprint* greater than the 80th percentile of all non-zero POH ≥ 80 % area *footprints*."

No, the 80th percentile is taken from the daily POH distribution (excluding footprints of size zero) of all days between April-September 2002-2019. The comparison with insurance loss data is done for the overlapping time frame 2003-2012.

10. P.5, I.118: Isolated hail days appear to require three non-hail days on either side, but not a single hail day in any 5-day window (a counter example are the two isolated southern hail days on days 176 and 180 in 2005 that are separated by only 3 days).

True. We now simplified the definition to just the second requirement (isolated hail days require three non-hail days on either side). See lines 127-128 (130-132):

"[...] and isolated hail periods to have only 1 hail day next to 3 non-hail days, both before and after. "

11. P.5, I.123: total hail day numbers do not match the 18-year annual averages given in ll. 113-114. Could this be due to seasonal bounds imposed?

Thank you very much for this observation. You are right, there was an error in counting the total number of hail days. The numbers in l. 113-114 had been correct and the numbers in l. 123 and Table 1 have now been corrected to a total of 476 hail days north of the Alps and 444 hail days south of the Alps. These corrected numbers agree with the 18-year average.

12. P.5, ll128-130: I am confused about how the independent cluster creation works.

Thinking it was too much detail, we didn't explain the cluster creation with all the necessary detail to reproduce it completely. The following example explains it fully.

Let us look at the two clustered periods in 2003, between doys 145-170 shown in the Table below. The algorithm has the purpose of splitting continuous clustering periods (like in A and B) into as many independent 5-day periods as possible. If at least 2 days separate 5-day clustering periods, then they can be considered independent. The blue 5-day periods with at least 4 hail days shown in Table R1 indicate all potential clustered 5-day periods. The periods in A cover 13 continuous "clustered" days and in B 7 days. In all cases in which 12 and up to 19 continuous days are part of potential 5-day clustering periods, the first and last 5 days are called "independent" and used for the subsequent analysis. This is done in A (shown in orange; ideally, the first 5-day period would have started one day later). In B, the clustering period lasts 7 days, that is less than 12. The easiest solution would have been to just pick the first 5 days. However, we felt it more sensible to try to pick days that are located centrally within these series of clustered hail days. The clustering period is determined through the following formula:

The clustering period goes from the Sth to the Tth day, with
S = round_down(n_days/2)-1
and
T = round_down(n_days/2)+3

In B that is
S = round_down(7/2)-1 = 2 ; T = round_down(7/2)+3 = 6; the independent cluster goes from the 2nd to the 6th day of the continuous clustered hail day period.

| | A | | | | | | | | | | | | | | | | B | | | | | | | | | |
|---|---|---|---|---|---|---|---|---|---|---|---|---|---|---|---|---|---|---|---|---|---|---|---|---|---|---|
| DOY | 145 | 146 | 147 | 148 | 149 | 150 | 151 | 152 | 153 | 154 | 155 | 156 | 157 | 158 | 159 | 160 | 161 | 162 | 163 | 164 | 165 | 166 | 167 | 168 | 169 | 170 |
| hail day | 0 | 0 | 0 | 1 | 1 | 1 | 1 | 1 | 1 | 1 | 1 | 0 | 1 | 1 | 1 | 0 | 0 | 0 | 1 | 1 | 1 | 1 | 0 | 1 | 0 | 0 |
| not clustered | | | | | | | | | | | | | | | | | | | | | | | | | | |
| clustered | | | | 1 | 1 | 1 | 1 | | | | | | | | | | | | 1 | 1 | 1 | 1 | | | | |
| | | | | 1 | 1 | 1 | 1 | 1 | | | | | | | | | | | 1 | 1 | 1 | 1 | | | | |
| | | | | | 1 | 1 | 1 | 1 | 1 | | | | | | | | | | | 1 | 1 | 1 | | 1 | | |
| | | | | | | 1 | 1 | 1 | 1 | 1 | | | | | | | | | | | | | | | | |
| | | | | | | | 1 | 1 | 1 | 1 | 1 | | | | | | | | | | | | | | | |
| | | | | | | | | 1 | 1 | 1 | 1 | | | | | | | | | | | | | | | |
| | | | | | | | | | 1 | 1 | 1 | | 1 | | | | | | | | | | | | | |
| | | | | | | | | | | 1 | 1 | | 1 | 1 | | | | | | | | | | | | |
| | | | | | | | | | | | 1 | | 1 | 1 | 1 | | | | | | | | | | | |
| duration of continuous clustered hail day period | | | | 13 days → | | | | | | | | | | | | | | 7 days → | | | | | | | | |
| independent clusters | | | | 1 | 1 | 1 | 1 | 1 | | | | | 1 | 1 | 1 | | | | 1 | 1 | 1 | 1 | | | | |

The algorithm then checks if this S to T 5-day period truly has at least 4 hail days and if that is not the case (because of three hail days between two single non hail days 011011101), it displaces the S to T period by -1 day. That case only happens once for the clustering period north of the Alps in the year 2004 (see Figure R4).

[Figure]

Figure R4: Same as Figure 2 in the article, but with the distinct information on which days are clustered hail days (dark dots) are which of these are within independent 5-day periods (dark circles).

Thinking back, The independent hail day clusters did not need to be only 5 days long; they could have lasted 6 or 7 days as well. Another criticism on this method is that some 5-day clusters contain only 4 hail days, even if 5 would have been possible by displacing the independent 5-day cluster period by just one day (see for example the first independent cluster period north of the Alps in 2009). However, the co-authors agreed to manually interfere as little as possible.

Just another comment on Figure 2 and Figure R4: For the clustered composites and significance tests, we only consider hail days within the independent 5-day clustering periods (shown in Figure R4 but not in Figure 2). We chose to rather not add the information about which days are within independent clustering periods in Figure 2, because it would likely confuse the reader. We would have to explain the independent cluster creation in more detail in the main paper as well. Knowing exactly which days are within the independent clusters is not crucial to understanding the rest of the paper. For this reason, we chose to only indicate which days are clustered hail days and which are not.

13. P.8, Fig.3: I recommend presenting the annual hail climatology by calendar month in some form as most readers would better related to a discussion / presentation that refers to hail frequency peaks in July rather than days 180- 199 or equivalent. I understand there are some minor inconsistencies in the mapping from DOY to months around the non-uniform duration of months (28-21 days) and the issue

around the four leap years in the dataset. Perhaps a secondary x-axis for non-leap years is one option?

The issue with leap-years is exactly the reason why we did not split the summer into months, but rather into 20-day sectors. To not create any more confusion about the method applied (using 20-day sectors), we chose to keep the bar-width of 20 days. However, we now added information on dates in both Figures 2 and 3. Figure 2 now includes a secondary x-axis indicating the first day of each month in a common year. The end of the Figure caption now has two more sentences: "The top x-axis indicates the first day of each month for common years. For leap-years, the axis needs to be displaced by one day to the right.".

The x axis labels in Figure 3 now includes for each 20-day sector the dates of a common year in the format "dd.mm - dd.mm" and the sentence of the figure caption sentence describing the x axis now says: "The numbers on the x-axis indicate the day of the year *and the equivalent dates of a common year*."

14. P.8, ll.166-167: This paragraph needs more explanation around the Weusthoff Weather Type Classification for interpretability. For example, is the weather type wind direction referring to the 500 hPa flow, the surface flow, some layer- mean wind direction, or other? Is there a geographical weighting on Central Europe, or all of Europe, or some other sub-region? The paragraph also needs more explanation of how the weather type across all isolated/clustered hail days was derived. Was the WTC first applied on a day-by-day bases, and then composited in some way?

The following Figure R5 (copied from Weusthoff et al. 2011) shows the exact domain within which the WTC is calculated.

[Figure]

*Figure 1: Classification domain.*

Figure R5: Domain used to calculate the weather type classification (copied from Weusthoff et al. 2011).

We first describe the WTC in section 2.2 and thus added some information there. This paragraph now says (lines 83-88 (84-91)):

"In addition to raw reanalysis data, this study uses an automated daily weather type classification (WTC) centered over the Alpine domain, which is based on the COST 733 Action methodology (Philipp et al. 2016) and run operationally at the Swiss weather service MeteoSwiss. Ten synoptic patterns are differentiated based on the geopotential height at 500hPa from ERA-Interim reanalysis (prior to 2011) and operational ECMWF Integrated Forecasting System (IFS) (from 2011). The ten classes correspond to the eight main wind directions and low- and high-pressure situations, respectively (see Weusthoff, 2011, Beck, 2007, for a full description)."

15. P.8, l.170: 'twice as frequent' when comparing ~28% to ~19% is a tad generous, how about 'significantly higher'?

True, thank you. That was an error. We changed lines 176-178 (180-183) to: "North of the Alps, westerly flow is more common during isolated hail days (43 % compared to 27 %), *northwesterly flow is as common and* northerly flow is more common during clustered hail days (22 % compared to 10 %). South of the Alps, the fraction of days with a northwesterly flow is *higher* for clustered hail days than for isolated ones *(28 % compared to 19 %)*."

16. P.10,ll.191-193: The sentence switches from dealing with the clustered day pattern to the isolated day pattern mid-sentence. It would be clearer to separate the two regimes through the structure in the text to avoid confusion.

We forgot to add a break in paragraph before the sentence starting with "Hence, " (see lines 199-201 (203-205)) and amended that. Is it sufficient?

17. P.10, l.197: Plot 5(b) does not show SSW 850 hPa flow over Switzerland given the flow magnitude is less than 4 m s-1 ; could this be a case of 'not shown'?

Yes, thank you, this is a case of "not shown". The sentence on lines 206-207 (211-212) now says:

"On clustered hail days, the winds are on average weaker than 4 m s-1 at 850 hPa (Fig. 5b) and flow from SSW over northern Switzerland (not shown)."

18. P.10, l.201: First *than* deserves deletion. Done.

19. P.11: The figure caption comes across as a tad organic. I would rewrite the caption by stepping through the individual panels (a), (b), ... and explain what is shaded and what is contoured. There is no need to mention against each individual panel that the sample processes consist of 135 clustered and 69 isolated hail days (mention that once only at the start or end).

Also considering the next review point 20, we rewrote the caption of Figure 5 as follows:

"Weather patterns during clustered and isolated hail days north of the Alps. (a) Potential vorticity at 335 K (color shading; in PVU), wind at 250 hPa (vectors, in m s-1) and atmospheric blocking frequency (dark green contours for 5 %, 7 %, and 10 %) on clustered hail days (n = 135 days), (b) TPW (filled contours, in mm) and wind at 850 hPa (vectors) on clustered hail days, (c and d) same as a and b but for isolated hail days (n = 69 days), (e) difference a minus c of PV at 335 K (color shading) and winds at 250 hPa (vectors). (f) difference b minus d of TPW (color shading) and winds at 850 hPa (vectors). (e and f) Statistically significant differences for > 50 % (lighter contour lines) and > 80 % of the resample composites (darker contour lines) are shown in e in green  for PV and in blue for wind speeds at 250 hPa and in f in green for TPW and in blue for wind speeds at 850 hPa. The contour lines indicating significances are smoothed using a 2D Gaussian filter (standard deviation = 1). The brackets below the wind vector legends indicate the minimum wind speed that is visualized. The grey contours show coastlines, and the black contour line shows the border of Switzerland."

20. P.12: The figure caption would benefit from a rewrite akin to the caption of Fig. 11. Details about the significance testing methodology don't need to be repeated in the caption. The CAPE field shown requires some clarification: I assume you are showing either SBCAPE or MUCAPE as a field, and given the XXCAPE value I shown for a hail day, similar to the 2-m temperature, and I assume panels (b) and (d) show the daily maximum XXCAPE value? The frequency of fronts being present is ambiguous and requires some more detail. If a front is detected by the Schemm scheme for 1 (2, or 3) of the 6 hour outputs from ERA-I, does the hail day count as a 'front day?' Panel f lacks explanations of the dashed orange and blue solid contours.

Thank you. Following your advice, the figure caption of Figure 6 was updated to the text below. The CAPE fields show the maximum MUCAPE of the day.

The Figure caption now says:

"Figure 6: Weather patterns during clustered and isolated hail days north of the Alps. (a) daily maximum T2M (color shading; in °C) and daily mean sea-level pressure (black contour lines, labels indicate by how much the MSLP exceeds 1000 hPa in hPa) on clustered hail days, (b) MUCAPE (color shading, J kg-1) and front frequencies (red, labelled lines) on clustered hail days, (c and d) same as a and b but for isolated hail days. (e) Difference a minus c of T2M (color shading) and MSLP (black contour lines). Statistically significant differences for more than 50 % of the resample composites (lighter contour lines) and more than 80 % (darker contour lines) are shown in green for T2M and in blue for MSLP and smoothed as in Fig. 5. (f) difference b minus d for MUCAPE (color shading) and front frequency (blue and red contour lines), the areas with > 50 \% significant differences in MUCAPE are shown in blue hashes (> 80 \% almost never present)."

About fronts: So far we have not defined days as being front days or not. The grid-boxes in which fronts are located each 6 hours are relatively few. Despite the

fronts visually displaying a relatively stationary behaviour, the concerned grid-boxes will likely still have changed from one 6-hour image to the next. Furthermore, we would have to declare how close and frequent a front needs to be to define a front day; that is quite a challenge. The contours in Figures 6, 10 and 13 indicate the front frequency for all 6-hourly time steps; unlike blocks, the fronts are not aggregated to daily values.

21. P.13, ll.218-219: A reference that 250 hPa diffluence sustains amplification of an upstream trough would be good here

We added the references (Shutts, 1983; Moore et al., 2008) in that sentence (see line 229 (236)).

22. P.13, l.219: What does it mean that the rough widens zonally? Given we are looking at a composite, this could mean that the meridional trough placement in the composite members becomes more diverse. Alternatively, it could mean that the trough is getting wider across the members.

When looking at single 6-hourly maps of PV, we notice that both are true. Sometimes the upstream trough is wider, sometimes the trough is zonally displaced. It can also happen that there is a cut-off at the trough location in the composite. In the present version of the article, this sentence has been removed (see lines 229 (234-236)):

"Such diffluence sustains the meridional amplification of the upstream trough (see e.g., Shutts, 1983; Moore et al., 2008)."

23. P.13 section 4.3.2: The section is not a strong contribution to the paper, apart from the gradual amplification of the pattern ahead of both types of hail days. A lot of the statement in this section are isolated single sentences that are not linked to a meaningful wider storyline. How do these isolated statements fit into a larger view of the lead-up to the cluster and isolated hail days?

We have edited this section to highlight the key points more clearly, the revised version now reads (lines 224-239 (229-247)):

"Composites of the days preceding the hail days illustrate the evolution of the upper-level changes that result in the more meridionally amplified flow over Europe on clustered hail days. These composites each consist of 31 days (see Table 1). For clustered hail days three days prior to the events north of the Alps, the trough over western Europe at 18°W already exists, and a ridge is present over central Europe (Fig. 7a). On at least 10% of the days, an atmospheric block is present over Scandinavia north of that ridge. As a consequence the flow is highly diffluent upstream of the ridge, and the zonal flow over Europe is very weak. Such diffluence sustains the meridional amplification of the upstream trough (see e.g., Shutts, 1983;

Moore et al., 2008). The moisture content of the atmosphere is relatively high (20-25 mm) three days before clustered hail day episodes (Fig. 8a) and increases by 2 mm from d-2 to d-1 (Fig. 8d and g). For isolated hail days the troughs and ridges over Europe also amplify from d-3 to d-1 (Fig. 7b, e and h). Atmospheric blocking is present over the western and central Atlantic but not over Europe (Fig. 7b, e and h). The ridge over Europe is weaker and the westerlies stronger compared to the clustered days already on d-3 to d-1. The moisture content is lower (17-22 mm on d-3) than before clustered hail days and increases by 2 mm from d-2 to d-1 (Fig. 8b, e and h).

In summary, the flow is more diffluent and meridionally amplified over Europe three days prior to clustered hail days compared to isolated hail days. Prior to both clustered and isolated hail days, the local atmospheric moisture content increases slightly from d-2 to d-1."

24. P.17,18: I recommend running a smoother over the significance testing sample fraction contours as these are very hard to read and interpret. Similar comments would apply to Fig. 5, 6.

Thank you. We did as suggested and applied a 2D Gaussian filter smoothing on the contours indicating significance in all the Plots. See Figures 5, 6, 9 and 10 and the adapted captions.

25. P.22, ll. 279-281: Sentence should start with an expression that indicates you have switched to joint cluster days

We added the following sentence at the beginning of the paragraph (lines 286-287 (294-295)):

"The next paragraph discusses the differences in weather pattern between clustered hail days that are clustered only south of the Alps compared to concurrent hail days both north and south of the Alps. "

26. P.23: Figure caption – 'fronts' is the proportion of the southern cluster (joint cluster) days with a diagnosed front?

It is the front frequency over all 6-hourly time steps during the overlapping days (joint cluster) and during the days clustering only south (southern cluster). Does that answer the question?

27. P.24, l.289: Better to use 'synoptic pattern' rather than 'situation'. Thank you.

28. P.24, l.299: This phrase seems to contradict the previous one where downstream anticyclonic Rossby wave breaking was postulated to result in weak upper-level and surface zonal flow over central and eastern Europe. In Fig. 5 the block also seems to

be sitting north of the UK and west of Scandinavia. These two sentences need a small amount of reconciliation.

Thank you for pointing out the unclear formulation. What we meant to say with the second sentence is that the weather systems move more slowly and are hence persistent from a Eularian point of view. We replaced the term "flow" with "weather systems" (lines 309-312 (318-321)).
"The flow is more amplified meridionally, and Rossby waves break downstream of Switzerland, resulting in weak upper-level zonal winds over central and eastern Europe and weaker surface zonal winds. In addition, blocking anticyclones over Scandinavia can contribute to more persistent *weather systems* over central Europe (Mohr et al., 2020)."

29. P.24,ll.300-304: These lines are quite speculative keeping in mind that the frontal analysis is based on a reanalysis and is composited. There could be a range of initiation mechanisms at work on cluster days that would require a different data set to resolve. Are 'thermo-topographic winds' related to upslope flow forced by elevated solar ground heating?

Yes we fully agree that the reanalysis datasets are too coarse to resolve the relevant flow structures. We have changed the formulation in the paper to clarify this point and to highlight that this point is indeed only a hypothesis. Yes, thermo-topographic winds are winds generated through differential heating in topography and include slope winds, valley winds and alpine pumping. (e.g. mountain breeze, alpine pumping).

We added the following sentence in lines 316-317 (325-326):
"These hypotheses would have to be confirmed using another data set that resolves these flow structures."

30. P.24, ll.307-308: There is an argument pointing in the opposite direction – more meridional patterns with weaker flow and reduced horizontal pressure gradients are more difficult to predict for numerical weather prediction models. I am guessing that you are saying that slowly evolving patterns are easier to predict through persistence arguments?

Yes, your guess was our line of thought. But we agree with your argumentation and therefore added the following sentence in lines 320-323 (329-332):

"On time scales of several days, the strongly meandering atmospheric flow and weaker pressure gradient may lead to clustered hail days being harder to predict than isolated hail days (Faranda et al., 2019). On a shorter time scale, the slow-moving large-scale flow signal suggests that clustered hail days might be more predictable than isolated hail days (e.g., Trapp, 2014; Dalcher and Kalnay, 1987)."

31. P.24, I.311-312: This argument is not as strong as it first sounds. Moist boundary layer air can be advected over distances much larger than 600 km and thus any local evapotranspiration to boost the local dewpoints is not required. For example, moisture-hungry supercells in Nebraska over 1000 k from the Gulf of Mexico are routinely supplied by such advected moisture when a lee trough east of the Rocky Mountains drives a stream of moist air northward. Local ET may help, but it is not a necessity based on a distance-to- coast argument.

While moisture may be advected from large distances, we have some evidence that moisture advection is less relevant than local sources for hail precipitation in northern Switzerland. An analysis (not yet published) of moisture sources of hail storms in northern Switzerland shows that on average 88% of the moisture has a land source and 20 to 30% of the land source is local. The hail storm of 6th June 2015 in northern Switzerland (Trefalt et al. 2019) picked up about a third of its moisture locally as well. This moisture had been supplied by precipitation the day before in the area that the low-level air parcels crossed. The numbers are different for southern Switzerland where the fraction of ocean moisture sources is higher. Reasons why the moisture transport from the sea may behave differently in northern Switzerland than in the US could be the latitude and differences in position relative to the oceans and the mountains.

32. P.25, L.324: Blocks are not obvious in the composite – they might still occur in individual cases

This part of the sentence was changed to (lines 339-340 (348-349)): "with the distinction that atmospheric blocks do not occur as frequently over northern Europe as for north of the Alps."

33. p.26, ll.372-375: The driver for convective initiation can not be derived from the data that have been used in this study and are supplied as an external input based on additional experience or datasets relating to a finer spatial scale.

Yes we agree with the reviewer that the reanalysis data sets are too coarse to resolve the convection triggering processes and we did not want to imply this. We have therefore changed the formulation of this sentence accordingly (see also Reply to the previous comment nr. 29).

We added a sentence in line 392 (401-402). The sentences on lines 389-392 (399-402) now say:
"Our findings suggest that on the north side of the Alps, thermo-topographic winds *may be* more relevant for convection initiation during clustered hail days, whereas prefrontal convergence and prefrontal orographic flow may be more important for the initiation of hailstorms on isolated hail days. *These hypotheses could be verified using a less coarse data set.*"

**3rd review**

Review of: "Multi-day hail clusters and isolated hail days in Switzerland – large-scale flow conditions and precursors", by Barras et al.

**Summary**

This study documents and compares the frequency of multi-day hail clusters versus isolated hail days in Switzerland. It shows that multi-day clusters are relatively common during summer, and that conditions relating to large, longwave troughs favor their occurrence.

I think this is solid research, and a well-designed and well-written paper. My comments are relatively minor, and mostly regard clarifications.

Thank you very much!

**Comments**

Figure 1: some lat/lon information on this figure would be useful (it's not clear to me what the km values relate to)

We replaced the coordinate system with WGS84 Lat/Lon scales and removed the last sentence "Switzerland is centered at ~46.8° N 8.2° E." in the Figure caption.

line 92: I assume that such a calculation of BWS is standard practice for this part of the world? Because of the complex topography, does using the 10 m winds i/o 850 hPa lead to excessively noisy BWS analyses?

Yes, because of the complex topography, we did not take the 10 m winds, because the fields are quite noisy. As you can see in Figure R2 and R3 answering Reviewer 2's main comment, the BWS shows unphysical values over the Alpine ridge as well.

line 105: Just to confirm that I understand this: the percentiles here are based on the set of daily areas over which POH>80%?

Yes, the calculation uses all days in which there is at least one grid-box with POH ≥ 80%. Any day with no POH ≥ 80% occurrence is not included in the percentile calculation.

See line 111-112 (114-116): "We tested minimum footprint area thresholds between the 70th and the 95th percentile of the daily non-zero footprint area distribution in the northern and southern domains."

line 108: This is a bit confusing/ I assume that 'nonhail days' here refers to days *not* identified by the POH footprint criteria? (rather than an "actual" nonhail day?) I'm trying to reconcile >5 car insurance hail losses in absence of hail.

Yes, non-hail days are days in which the footprint criterion is not met.
We clarified it in the text with a bracket in line 116 (119-120):

"non-hail days *(days in which the footprint percentile threshold is not exceeded)*"

line 118: While I appreciate the value in following the published approach of other researchers, I think it's worth giving a very brief justification (meteorological or otherwise) on the significance of 5-day periods. To be clear, I think I can infer the justification, and I don't have an issue with this period of consideration, but it is fair to ask why 5 is preferable.

Agreed. In summer 5 days is a time period where we expect the local weather to be influenced by a single synoptic system. The choice of five day periods rather than four day periods is somewhat arbitrary. We added a sentence on lines 126-127 (129-130):
"To define the clustered hail periods and isolated hail days, we use a counting approach similar to Pinto et al. (2014) and Kopp et al. (2021). *We also consider that in summer, 4 or 5 days is a typical time duration in which a single synoptic system influences the local weather.*"

line 150: s/b 'resampled'. Thank you.

line 363: Using your analysis, you could also express this in terms of a probability: Given a hail day, there is an xx % probability that this day is part of a multi-day cluster.

Good suggestion. Since the year to year variability is so high, expressing it in terms of averages seems more transparent, so we left it as it is.

line 388 (future research): One of the cited North American studies speculated that upscale feedbacks driven by the diabatic heating of deep convection may contribute to the consecutive days of hazardous convection. Not knowing the total size and duration of the convective storms contributing to clustering, it's difficult to know whether this idea has any relevance to Switzerland, but it's perhaps something to consider. If relevant, this would have implications, for example, on the necessary approaches to climate modeling (resolution, global versus limited-area, convective parameterizations, etc.).

Thank you for this interesting perspective. In fact, we are planning to analyse the characteristics of individual storms in the future; for this your comment is very relevant. We adapted the respective paragraph to clarify that also the area and intensity of the storms are relevant factors to look at.

**Literature:**

Joe, P., Burgess, D., Potts, R., Keenan, T., Stumpf, G., and Treloar, A.: The S2K severe weather detection algorithms and their performance, Weather and Forecasting, 19, 43-63, https://doi.org/10.1175/1520-0434(2004)019%3C0043:TSSWDA%3E2.0.CO;2 , 2004.

McGill, R., Tukey, J. W., and Larsen, W. A.: Variations of box plots, The American Statistician, 32, 12-16, https://doi.org/10.1080/00031305.1978.10479236 , 1978.

Krzywinski, M. and Altman, N.: Visualizing samples with box plots, Nat Methods, 11, 119-120, https://doi.org/10.1038/nmeth.2813 , 2014.

Schemm, S., Nisi, L., Martinov, A., Leuenberger, D., and Martius, O.: On the link between cold fronts and hail in Switzerland, Atmospheric Science Letters, 17, 315-325, https://doi.org/10.1002/asl.660 , 2016.

Trefalt, S.: Hail and Severe Wind Gusts in the Convective Season in Switzerland, Ph.D. thesis, University of Bern, 2017.

Trefalt, S., Martynov, A., Barras, H., Besic, N., Hering, A. M., Lenggenhager, S., Noti, P., Röothlisberger, M., Schemm, S., Germann, U., and Martius, O.: A severe hail storm in complex topography in Switzerland - Observations and processes, Atmospheric Research, 209, 76-94, https://doi.org/10.1016/j.atmosres.2018.03.007 , 2018.

Treloar, A.: Vertically integrated radar reflectivity as an indicator of hail size in the greater Sydney region of Australia, in: Preprints, 19th Conf. on Severe Local Storms, Minneapolis, MN, Amer. Meteor. Soc, pp. 48-51, 1998.

(see also main article)

---

## Referee Report (RR1)

**Second Review of WCD-2021-25**

**Title:** Multi-day hail clusters and isolated hail days in Switzerland –
large-scale flow conditions and precursors

**Authors:** Hélène Barras, Olivia Martius, Luca Nisi, Katharina Schroeer, Alessandro Hering, and Urs Germann

**Recommendation:** Minor revision

I very much appreciate the significant effort that the authors have invested to address all the comments made by the three reviewers. Such efforts show the desire to arrive at a stronger publication, and the manuscript at hand is no exception. I have a few broader considerations that the authors might to address beyond their initial response, and I consider these as minor overall.

The response to the wind shear query of my first review mentioned that a subset of the hailstorms captured in the hail cases for this study were from supercells. On conceptual grounds these storms require deep layer shear which may or may not be captured adequately on the 0.5º ERA-5 dataset in the vicinity of substantial terrain (which would not be captured adequately on 0.5º grid). I see a risk of a proportion of the readership might gain a similar impression, which can preemptively be addressed somewhat more that it has to date. I recommend the following considerations for inclusion.

[1] Hail of golf ball size or larger is most likely due to storms that are organized due to deep layer shear interaction (what proportion of the hail reports in the insurance dataset are in that category?)

[2] Can anything be said about the deep layer performance of ERA-5 around significant orography (I am not asking for this step, but has anyone ever plotted observed or km-scale modelled deep layer shear against ERA-X deep layer shear?)

The restate the above thoughts once more has they summarize the main gap in demonstrating that ERA-5 can be meaningfully applied to flows around the Alps in the lower troposphere.

Among the very minor tidy-up considerations are:

- Clustering hail days: I can see that the amount of detail required to explain the approach can be seen to detract from the paper's main messages. On the other hand, a study should provide the minimum amount of information that allows

a (rather keen) scientist to reproduce the results. Maybe the best compromise is an Appendix with the clustering details, or a reference to an external source? The allocation of two 5-day clusters to 12-19 day periods of clustered days seems to follow the principle of maximum 5-day period separation. This principle can lead to the 5-day periods extending beyond the actual hail days by 1 day. Is there a simple way of showing that integrating such non-hail days into the analysis is not going to majorly alter the results? I suspect this may be hard to show, and I do not consider it an essential inclusion.

---

## Author Response (AR2)

Many thanks to Harald Richter for the valuable additional comments and questions. Based on these, we were able to further improve our article. The comments point out that more research is needed, for instance to determine the quality in wind shear simulations in complex topography in reanalyses. We hope that the answers and changes in the article are satisfactory.

**Second Review of WCD-2021-25**

**Title:** Multi-day hail clusters and isolated hail days in Switzerland –
large-scale flow conditions and precursors
**Authors:** Hélène Barras, Olivia Martius, Luca Nisi, Katharina Schroeer, Alessandro Hering, and Urs Germann
**Recommendation:** Minor revision

I very much appreciate the significant effort that the authors have invested to address all the comments made by the three reviewers. Such efforts show the desire to arrive at a stronger publication, and the manuscript at hand is no exception. I have a few broader considerations that the authors might to address beyond their initial response, and I consider these as minor overall.

The response to the wind shear query of my first review mentioned that a subset of the hailstorms captured in the hail cases for this study were from supercells. On conceptual grounds these storms require deep layer shear which may or may not be captured adequately on the $0.5^o$ ERA-5 dataset in the vicinity of substantial terrain (which would not be captured adequately on $0.5^o$ grid). I see a risk of a proportion of the readership might gain a similar impression, which can preemptively be addressed somewhat more that it has to date. I recommend the following considerations for inclusion.

[1] Hail of golf ball size or larger is most likely due to storms that are organized due to deep layer shear interaction (what proportion of the hail reports in the insurance dataset are in that category?)
Unfortunately we do not have information on the hail size from the insurance reports, we only know that damages to cars usually only occur for hail sizes > 2 cm.

[2] Can anything be said about the deep layer performance of ERA-5 around significant orography (I am not asking for this step, but has anyone ever plotted observed or km-scale modelled deep layer shear against ERA-X deep layer shear?)

We are not aware of any validation of ERA-X deep layer shear against observations in complex orography. Taszarek et al. 2021 compared wind shear in ERA-5 with wind shear derived from rawinsondes in Europe and North America and found that ERA-5 tends to underestimate wind shears and that especially extreme wind shear (>=25ms-1) are not well represented. There is a study by Graf et al. 2019 validating regional climate model winds against observations and these comparisons clearly show that the regional climate model simulations fail to capture the diurnal wind systems in valley locations. These thermo-topographic wind systems are important for the formation of shear in Switzerland and in the pre-Alps on days with severe convection (see e.g., Trefalt et al. 2018). It may be that the assimilation of observations into ERA-5 solves this issue, but we do not know. We therefore include a sentence in the summary and discussion section stating that the winds in ERA-5 close to complex topography may not include important thermo-topographic wind systems due to the coarse model resolution.

We added the following sentence in lines 369-372:

Thermo-topographic wind systems, such as diurnal winds in valleys, are important for the formation of wind shear in the pre-Alps on days with severe convection (see e.g. Trefalt et al., 2018). These winds are potentially insufficiently resolved by the coarse-resolution reanalysis, which could explain this lack of difference between clustered and isolated hail days.

The restate the above thoughts once more has they summarize the main gap in demonstrating that ERA-5 can be meaningfully applied to flows around the Alps in the lower troposphere.

Among the very minor tidy-up considerations are:

• Clustering hail days: I can see that the amount of detail required to explain the approach can be seen to detract from the paper's main messages. On the other hand, a study should provide the minimum amount of information that allows a (rather keen) scientist to reproduce the results. Maybe the best compromise is an Appendix with the clustering details, or a reference to an external source? The allocation of two 5-day clusters to 12-19 day periods of clustered days seems to follow the principle of maximum 5-day period separation. This principle can lead to the 5-day periods extending beyond the actual hail days by 1 day. Is there a simple way of showing that integrating such non-hail days into the analysis is not going to majorly alter the results? I suspect this may be hard to show, and I do not consider it an essential inclusion.

Thank you for your suggestion. We now added another Appendix chapter that explains the definition of independent 5-day clustered hail day periods, similarly to the review answers (see below). To answer the last question: The independent 5-day clustered periods sometimes include non-hail days (4 hail days + 1 non-hail day), but these non-hail days are not included in the hail-day analyses at all. These non-hail days are only considered in the analysis if they precede the hail-day clusters and, therefore, are counted as non-hail days before clustered hail days ("d-1" days in Table 1).

The second paragraph in Appendix C now says (see lines 431-451):

The clustered hail days are by nature dependent. We therefore apply a 500-times-repeated resampling to all clustered and isolated hail days such that each of the 500 series contains only serially independent data. Isolated hail days are by nature independent; this category does not need any additional treatment to ensure independence. However, continuous periods with clustered hail days should only be sampled once per resampling. To split all periods of clustered hail days into independent 5-day clustering periods, we apply an algorithm. This algorithm treats clustered periods lasting up to 11 days differently than clustered periods longer than 11 days. In the latter case, the > 11-day-period is divided into periods of 5 days that have at least 2 days between each other. Concretely, the first and last 5 days defined as clustered hail days are chosen (see e.g. DOYs 147–159 in the year 2003). The 5-day period in ≤ 11-day situations is set to go from the Xth to the Yth day, with X and Y being defined as follows:

$$X = \mathrm{floor}(n/2) - 1$$

$$Y = \mathrm{floor}(n/2) + 3$$

with n being the number of days in a period of clustered hail days and "floor" denoting that the result of the fraction is rounded down to the next integer. If a period of clustered hail days lasts e.g. 7 days, then the 5-day period we call independent will start on the second and end on the sixth day. For the clustering period in 2004 (Fig. 2), some clustered hail days have the sequence no hail (0) and hail days (1) "11011101". In this case, the 5-day period chosen by the algorithm

contains only 3 hail days (01110), despite being marked as clustered by their attribution to neighboring 5-day periods. In this case the choice is corrected by displacing the 5-day period to one day earlier. Consequently, the number of hail days per independent clustering period is always ≥ 4. This criterion of independence has the consequence of not including all potentially available clustered hail days. North and south of the Alps, this treatment additionally removes 29 and 13 out of 164 and 102 clustered hail days, respectively.

References:

Taszarek, M., Pilguj, N., Allen, J. T., Gensini, V., Brooks, H. E., and Szuster, P.: Comparison of Convective Parameters Derived from ERA5 and MERRA-2 with Rawinsonde Data over Europe and North America, Journal of Climate, 34(8), 3211-3237, https://doi.org/10.1175/JCLI-D-20-0484.1, 2021.